# A conserved CCM complex promotes apoptosis non-autonomously by regulating zinc homeostasis

Eric M. Chapman[1,2], Benjamin Lant[2], Yota Ohashi[1], Bin Yu[2], Michael Schertzberg[3], Christopher Go[1,4], Deepika Dogra[5], Janne Koskimäki[6], Romuald Girard[6], Yan Li [7], Andrew G. Fraser[1,3], Issam A. Awad[6], Salim Abdelilah-Seyfried [5], Anne-Claude Gingras [1,4] & W. Brent Derry [1,2]

Apoptotic death of cells damaged by genotoxic stress requires regulatory input from surrounding tissues. The *C. elegans* scaffold protein KRI-1, ortholog of mammalian KRIT1/CCM1, permits DNA damage-induced apoptosis of cells in the germline by an unknown cell non-autonomous mechanism. We reveal that KRI-1 exists in a complex with CCM-2 in the intestine to negatively regulate the ERK-5/MAPK pathway. This allows the KLF-3 transcription factor to facilitate expression of the SLC39 zinc transporter gene *zipt-2.3*, which functions to sequester zinc in the intestine. Ablation of KRI-1 results in reduced zinc sequestration in the intestine, inhibition of IR-induced MPK-1/ERK1 activation, and apoptosis in the germline. Zinc localization is also perturbed in the vasculature of *krit1*[−/−] zebrafish, and *SLC39* zinc transporters are mis-expressed in Cerebral Cavernous Malformations (CCM) patient tissues. This study provides new insights into the regulation of apoptosis by cross-tissue communication, and suggests a link between zinc localization and CCM disease.

[1] Department of Molecular Genetics, University of Toronto, Toronto, M5S 1A8 ON, Canada. [2] Developmental and Stem Cell Biology Program, Peter Gilgan Centre for Research and Learning, The Hospital for Sick Children, Toronto, M5G 0A4 ON, Canada. [3] The Donnelly Centre for Cellular and Biomolecular Research, University of Toronto, Toronto, M5S 3E1 ON, Canada. [4] Lunenfeld-Tanenbaum Research Institute at Mount Sinai Hospital, Toronto, M5G 1X5 ON, Canada. [5] Institute for Biochemistry and Biology, Potsdam University, Potsdam, 14476, Germany. [6] Neurovascular Surgery Program, Section of Neurosurgery, The University of Chicago Medicine, Chicago, IL 60637, USA. [7] University of Chicago Center for Research Informatics, The University of Chicago, Chicago, IL 60637, USA. Correspondence and requests for materials should be addressed to W.B.D. (email: brent.derry@sickkids.ca)

Apoptosis is a genetically regulated program that results in the removal of cells from a population in response to developmental cues or external stimuli[1]. In the nematode worm *Caenorhabditis elegans*, apoptosis occurs in the somatic tissue during development[2], and in the germline under normal physiological conditions or in response to DNA damage[3,4]. Apoptosis in the soma and in response to germline DNA damage is initiated by the expression of the pro-apoptotic BH3-only gene *egl-1* that activates a conserved signaling cascade. EGL-1 protein binds and inhibits the anti-apoptotic CED-9/BCL2, allowing for CED-4/APAF1 apoptosome formation and activation of the caspase CED-3[5]. Excessive DNA damage in the germline as a result of ionizing radiation (IR), activates the p53-like transcription factor CEP-1[6,7], which induces expression of the BH3-only gene *egl-1* to stimulate apoptosis[8]. In addition, the ERK1/2 homologue MPK-1 cooperates with the CEP-1 pathway[9] and can also function in parallel[10,11] to promote IR-induced apoptosis.

Previously, we found that KRI-1 is required in the soma to promote IR-induced germline apoptosis in a cell non-autonomous manner, independent of CEP-1/p53[12]. KRI-1 is the ortholog of the human scaffolding protein KRIT1/CCM1, which is frequently mutated in familial occurrences of the vascular disease cerebral cavernous malformations (CCM) that can lead to strokes and seizures[13]. Since KRI-1 is expressed in the soma, but not the germline[12,14] we wondered how this scaffold non-autonomously promotes apoptosis of damaged germ cells.

In this study, we define the apoptotic pathway downstream of KRI-1 by conducting a forward genetic mutagenesis screen to restore IR-induced apoptosis in *kri-1* mutant worms. This screen uncovers a conserved ERK-5/MAPK signaling pathway and the KLF-3 transcription factor[15,16]. Proteomic analysis of KRI-1 binding partners identifies the previously uncharacterized protein K07A9.3, which is the ortholog of human CCM2 and a regulator of the ERK-5 pathway. Transcriptome analysis in *kri-1* mutant worms reveals that the zinc transporter gene *zipt-2.3* is regulated by KLF-3 and required for intestinal zinc storage. This permits MPK-1 phosphorylation in the germline to promote apoptosis. By understanding the conserved KRTI1/CCM1 signaling network it will be possible to identify novel therapeutics for a condition that currently relies on invasive surgery for treatment[17].

## Results

**KRI-1 regulates MPK-1 to promote IR-induced apoptosis.** Given that the ERK1 homologue MPK-1 is necessary in the pachytene region of the germline for IR-induced apoptosis[9,10], we wondered if germline MPK-1 might be regulated by KRI-1. To test this, germlines were isolated from wild type and *kri-1 (ok1251)* null worms[12,14] and immunostained with an antibody that recognizes the active di-phosphorylated (dp-) MPK-1[18]. While levels of dp-MPK-1 were similar in wild type and *kri-1 (ok1251)* animals in non-irradiated conditions, *kri-1(ok1251)* worms had reduced levels of IR-induced dp-MPK-1 in the pachytene region compared to wild type animals (Fig. 1a). This indicates that KRI-1 in the soma is required for the full activation of MPK-1 in the germline. To determine whether increasing MPK-1 activation can restore apoptosis in *kri-1(ok1251)* mutants, we ablated two different negative regulators of MPK-1. First, we crossed a loss of function mutation (*zh10*) of the MPK-1 germline phosphatase *lip-1*[9] into *kri-1(ok1251)* mutants and found that the *kri-1(ok1251); lip-1(zh10)* double mutants had levels of apoptosis similar to *lip-1(zh15)* single mutants (Fig. 1b). Knock-down of the MPK-1 negative regulator *gla-3*[19] by RNAi in *kri-1(ok1251)* mutants also restored IR-induced apoptosis to similar levels as knocking-down *gla-3* in wild type animals (Supplementary Fig. 1A). To determine where *kri-1* intersects the MAPK pathway,

we crossed *kri-1(ok1251)* worms with a strain harboring a *let-60/Ras* gain-of-function mutation (*ga89*). This allele causes hyper-activation of MPK-1 in the germline[18,20] and increased sensitivity to IR-induced apoptosis[9]. The levels of apoptosis in the *kri-1 (ok1251); let-60(ga89)* double mutant was not as high as *let-60 (ga89)* single mutants (Supplementary Fig. 1B), indicating that KRI-1 promotes the activation of MPK-1 either downstream or independently of LET-60/Ras (Fig. 1c).

**The ERK-5 pathway and KLF-3 function downstream of KRI-1.** To identify genes that function downstream of *kri-1*, we performed a forward genetic suppressor screen using the mutagen ethyl methanesulfonate (EMS). Since scoring apoptosis in single worms under a compound light microscope is rate-limiting, a rapid screening method was required. We therefore exploited the hypersensitivity of *kri-1(ok1251)* mutants to starvation stress in order to select for EMS-induced mutations that suppress this phenotype (Fig. 2a), reasoning that these suppressors might also restore IR-induced germline apoptosis. From an initial selection screen involving approximately 1,000,000 haploid genomes, we isolated 300 second generation (F2) candidate suppressors that survived this period of diapause arrest (Supplementary Fig. 2A). Clonal populations from these survivors were established, and 13 of these suppressor strains isolated from unique populations exhibited a complete restoration of IR-induced apoptosis (Fig. 2b). We then performed whole genome sequencing to identify the EMS-induced mutations in the 13 *kri-1(ok1251)* suppressor strains. Dominance tests revealed that 12 strains had recessive mutations, while one strain harbored a dominant mutation (Supplementary Fig. 2B). Strikingly, the 12 recessive strains had non-synonymous mutations within genes that comprise the ERK-5/MAPK pathway (Fig. 2c, e). In this conserved pathway, the MAP kinase kinase kinase MEKK3 phosphorylates and activates the MAP kinase kinase MEK5 that then activates the MAP kinase ERK5[21]. Six strains had mutations that result in single amino acid changes in *Y106G6A.1*, which is orthologous to human *MEKK3* (Supplementary Fig. 2C). Of these six strains, two had unique point mutations, while four strains shared a third point mutation (Fig. 2c). Five strains had unique mutations in isoform "a" of the gene *E02D9.1*, which is orthologous to human *MEK5* (Supplementary Fig. 2D). Three of these *E02D9.1* mutations result in amino acid changes, one alters a splice donor site and one is a single amino acid deletion resulting in a predicted frame-shift (Fig. 2c). Finally, one strain had a mutation in a splice acceptor site in *mpk-2* (Fig. 2c), which is orthologous to human *ERK5* (Supplementary Fig. 2E). We henceforth refer to *Y106G6A.1* as *mekk-3*, *E02D9.1* as *mek-5*.

The single *kri-1* suppressor strain with the dominant mutation (Supplementary Fig. 2B) had a non-synonymous nucleotide change in *klf-3*, which encodes a "Group 2-like" KLF transcription factor[22]. This point mutation results in a glycine to valine substitution in the second C2H2 zinc finger-DNA binding domain of KLF-3 (Fig. 2d), and is an expected gain-of-function allele since haploinsufficiency is very rare in *C. elegans*[23]. Since the vertebrate ERK5 pathway is known to regulate KLF transcription factors in the context of CCM disease[16,24], it is likely that *klf-3* is part of the KRI-1/KRIT1 pathway in *C. elegans*.

**The ERK-5 pathway and KLF-3 regulate IR-induced apoptosis.** To validate that the predicted loss-of-function mutations in *mekk-3*, *mek-5*, and *mpk-2* cause a restoration of apoptosis downstream of *kri-1*, we knocked down each of these genes by RNAi in wild type and *kri-1(ok1251)* worms. Ablation of all three genes fully restored IR-induced apoptosis in *kri-1(ok1251)* animals, while no further increase was observed in wild type worms

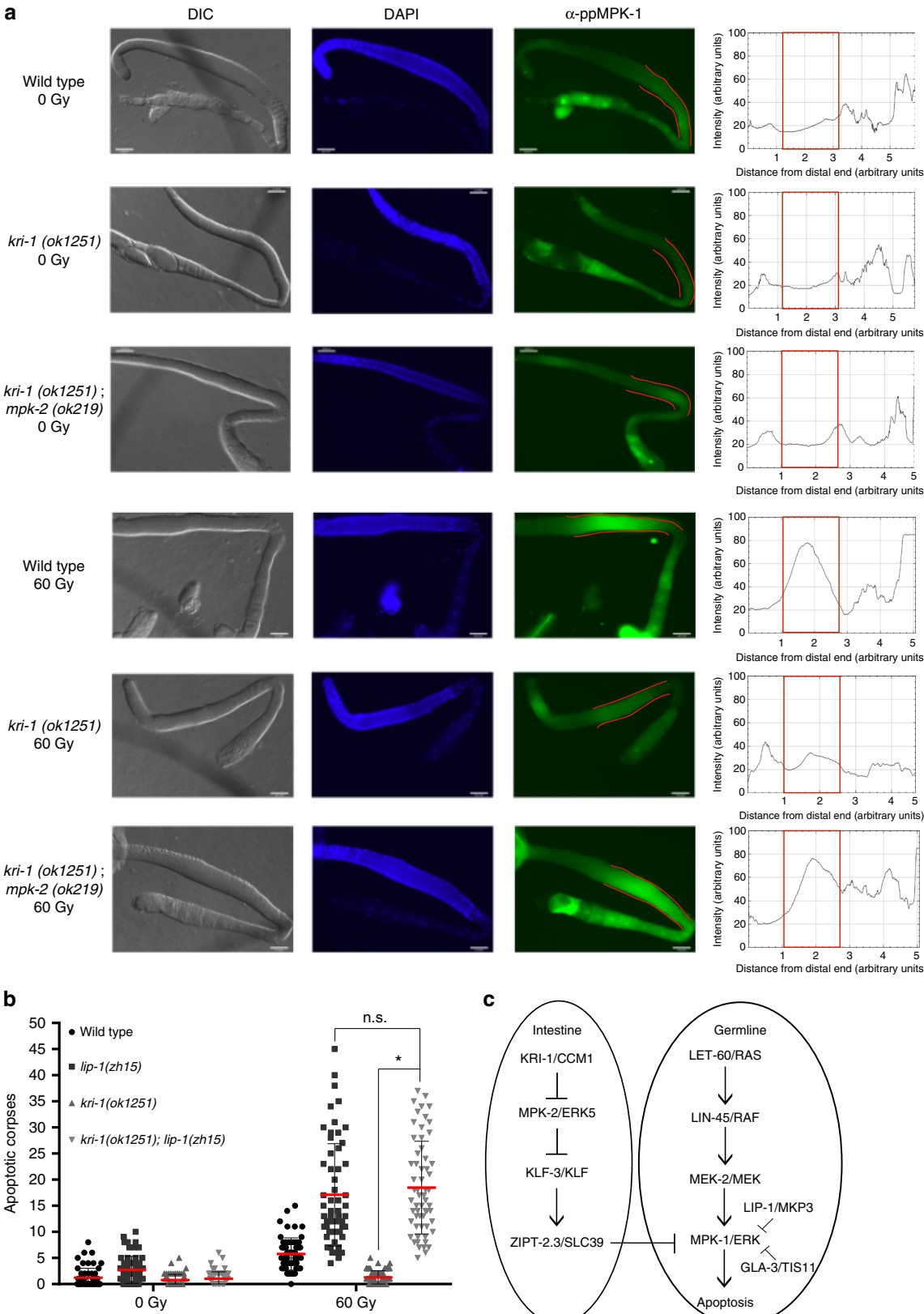

**Fig. 1** KRI-1 regulates MPK-1/ERK1 in the germline. **a** Di-phosphorylated MPK-1 in the germlines of wild type, *kri-1(ok1251)*, and *kri-1(ok1251); mpk-2(ok219)* animals is measured along the midline from the distal to proximal end. The red box on each graph corresponds to the pachytene region. Images are representative of at least 30 worms per strain, per condition (×400 magnification). Scale bar is 25 μm. **b** IR-induced germline apoptosis scored in wild type, *lip-1(zh15)*, *kri-1(ok1251)*, and *kri-1(ok1251); lip-1(zh15)* animals (*n* ≥ 60). Red line is mean ± standard deviation. *\*P* < 0.05, two-sided, unpaired *t*-test. **c** Schematic of intestinal KRI-1 regulating MPK-1/ERK1 in the germline

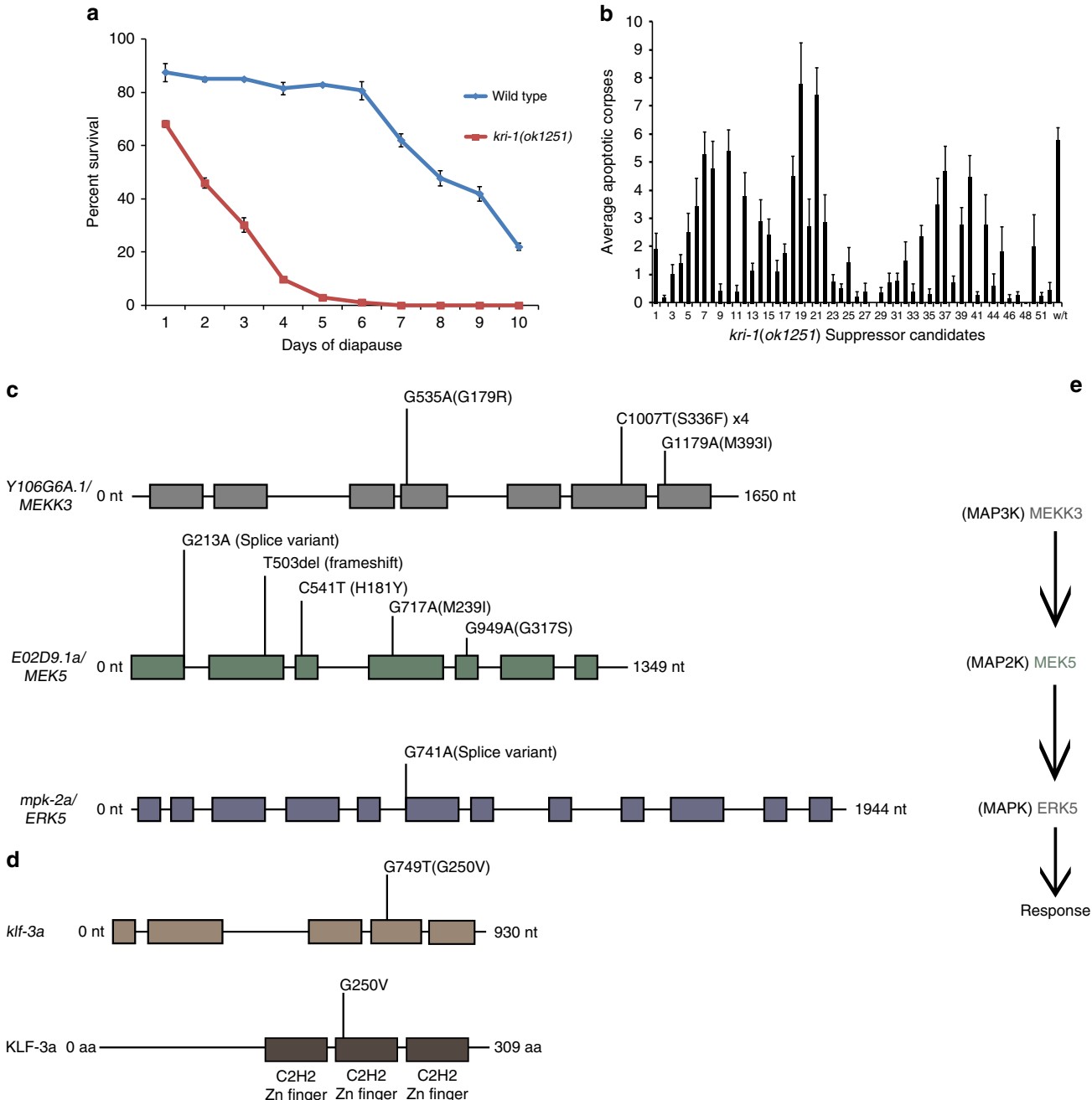

**Fig. 2** A *kri-1* suppressor screen identifies the ERK-5/MAPK pathway and KLF-3. **a** Survival of wild type and *kri-1(ok1251)* L1 larvae in liquid buffer (M9) without food ($n \geq 600$). **b** IR-induced germline apoptosis in the progeny of *kri-1(ok1251)* suppressor candidates that survived 7 days of starvation (52 of 300 candidates depicted, $n \geq 15$). The black bar is the mean ± standard deviation. **c** Twelve of the 13 *kri-1(ok1251)* suppressor candidates have recessive mutations in ERK-5/MAPK pathway genes. **d** The 13th *kri-1(ok1251)* suppressor candidate has a dominant mutation in the KLF transcription factor gene *klf-3*. **e** Depiction of the ERK-5/MAPK pathway

(Fig. 3a). To determine if this ERK-5 pathway functions as a general regulator of IR-induced apoptosis, we ablated each gene in *daf-2(e1370)* mutants (Fig. 3a), which are resistant to IR-induced apoptosis[11] by a mechanism that is independent of *kri-1*[12]. Since knockdown of these genes was unable to restore apoptosis in *daf-2(e1370)* animals, it is likely that the ERK-5/MAPK pathway acts downstream of KRI-1 to regulate IR-induced apoptosis. Since *mekk-3* and *mek-5* are very close to *kri-1* on chromosome I, we decided to focus on the terminal MAPK gene, *mpk-2* located on chromosome II for compound mutant analysis. We took advantage of the previously uncharacterized *mpk-2(ok219)* deletion allele, which removes the 5′UTR and the first

three exons of *mpk-2* (Supplementary Fig. 3A). Using this predicted null allele, we created a *kri-1(ok1251); mpk-2(ok219)* double mutant and found that IR-induced dp-MPK-1 (Fig. 1a) and apoptosis (Supplementary Fig. 3B) were restored to wild type levels. These results demonstrate that resistance to IR-induced apoptosis in *kri-1(ok1251)* animals is due to over-activation of the ERK-5/MAPK pathway, and that suppression of this cascade restores sensitivity. Finally, to determine if the mutation in *klf-3* was responsible for restoring apoptosis in the dominant *kri-1(ok1251)* suppressor strain, we generated the same *klf-3* point mutation by CRISPR/Cas9, and confirmed that this new *klf-3(on34)* allele functions in a dominant manner to restore

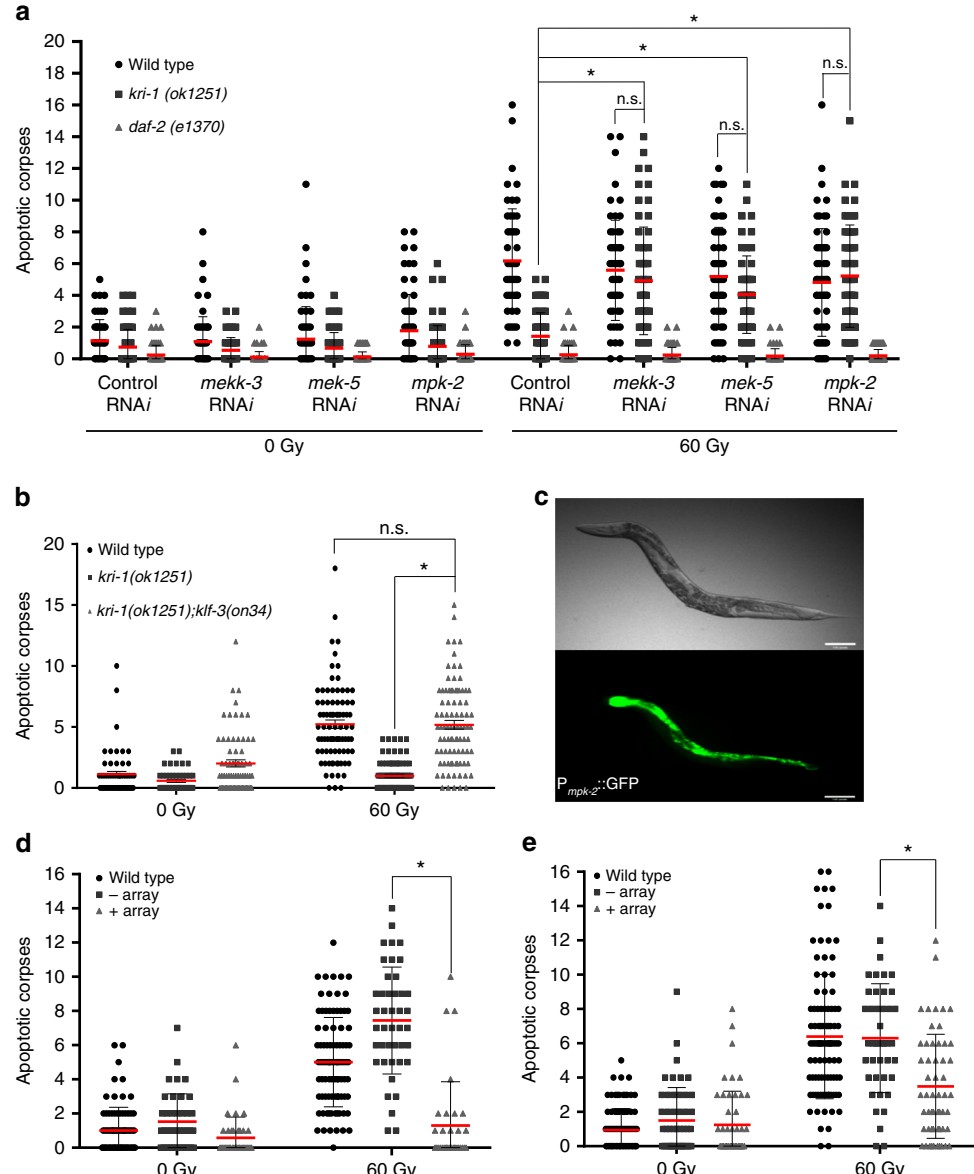

**Fig. 3** MPK-2/ERK5 and KLF-3 regulate IR-induced apoptosis downstream of KRI-1. **a** IR-induced germline apoptosis scored after knock-down of the ERK-5/MAPK pathway genes, *mekk-3, mek-5,* and *mpk-2* in wild type, *kri-1(ok1251)* and *daf-2(e1370)* animals ($n \geq 60$). **b** IR-induced germline apoptosis scored in wild type, *kri-1(ok1251)*, and *kri-1(ok1251); klf-3(on34)* animals ($n \geq 45$). **c** Expression of P*mpk-2*::*gfp*. Images are representative of three independent lines (×100 magnification). Scale bar is 50 μm. **d** IR-induced germline apoptosis scored in wild type, P*mpk-2*::*mpk-2* "+array", and P*mpk-2*::*mpk-2* siblings that have lost the transgene "−array" ($n \geq 30$). **e** IR-induced germline apoptosis scored in wild type, P*elt-2*::*mpk-2* "+array", and P*elt-2*::*mpk-2* siblings that have lost the transgene "−array" ($n \geq 30$). **a–e** Red line is mean ± standard deviation. *$P < 0.05$, two-sided, unpaired *t*-test

IR-induced apoptosis in *kri-1(ok1251)* mutants (Fig. 3b; Supplementary Fig. 3C). We conclude that this mutation confers a gain-of-function to KLF-3, which acts downstream of *kri-1* and likely the ERK-5/MAPK cascade.

**Intestinal ERK-5 regulates MPK-1 and IR-induced apoptosis.** To determine where *mpk-2* is expressed, we created a transcriptional reporter consisting of 6 kb of the region upstream of the *mpk-2* start codon, fused to GFP. This construct was injected into wild type worms to form extrachromosomal (Ex) arrays[25]. Three independent array-containing lines express GFP in the intestine (Fig. 3c) throughout development (Supplementary Fig. 3D), similar to KRI-1::GFP[14]. Since our genetic analysis predicts that MPK-2 is over-activated in *kri-1(ok1251)* worms, we over-

expressed *mpk-2* in wild type animals and quantified IR-induced apoptosis. Three independent Ex array-containing lines with *mpk-2* under the control of a 6 kb upstream element were expressed in wild type animals, and suppression of IR-induced apoptosis was observed compared to siblings that had lost the arrays (Fig. 3d). Since Ex arrays are silenced in the germline[26], this suggests that MPK-2 functions in the soma to inhibit germline apoptosis. To determine whether over-expression of *mpk-2* specifically in the intestine can confer resistance to IR-induced germline apoptosis, we expressed *mpk-2* under 5 kb of the *elt-2* intestinal-specific promoter from three independent Ex arrays in wild type worms. These strains had a reduction in IR-induced apoptosis compared to siblings that had lost the array (Fig. 3e), confirming that MPK-2 regulates germline apoptosis from the intestine. In addition, *kri-1(ok1251); mpk-2(ok219)*

double mutants have restored IR-induced dp-MPK-1 activation in the germline (Fig. 1a). We conclude that KRI-1 functions to suppress ERK-5/MAPK signaling in the intestine to facilitate the activation of germline MPK-1 and apoptosis in response to genotoxic stress.

**CCM-2 and ICAP-1 bind KRI-1 to promote IR-induced apoptosis.** Mammalian KRIT1/CCM1 regulates the ERK5/ MAPK pathway through its interaction with CCM2, which directly binds and inhibits the activation of MEKK3[27,28]. In *C. elegans*, a CCM2 ortholog could not be identified by sequence homology-based searches, so we employed affinity purification-mass spectrometry (AP-MS) to help illuminate the mechanism by which KRI-1 regulates the ERK-5/MAPK cascade. We took advantage of a previously constructed KRI-1::GFP strain[14] that restores IR-induced apoptosis in *kri-1(ok1251)* mutants[12]. A strain that expresses actin (ACT-5) fused to GFP in the intestine[29] and wild type worms that do not express GFP were used as controls. Anti-GFP antibodies were used to immunoprecipitate whole worm lysate from irradiated and non-irradiated samples. Only four proteins were found to interact with KRI-1::GFP regardless of irradiation status (Y45F10D.10, K07A9.3, F37C4.5, and HIP-1), while HSP-17 was found to interact with KRI-1::GFP exclusively post irradiation (Fig. 4a, b).

F37C4.5 is an uncharacterized *C. elegans* open reading frame with no clear homologues, while HIP-1 and HSP-17 are heat shock proteins. Y45F10D.10 exhibits 25% amino acid identity to the mammalian KRIT1 binding partner, ICAP1[30] (Supplementary Fig. 4A), and consistent with this, its predicted structure is best threaded with human ICAP1[31] as a template (Phyre2 confidence score 100.0; Figure S4A). The best structural templates for threading the amino and carboxy-terminal portions of K07A9.3 are two partial structures of human CCM2, namely its PTB domain[32] (Phyre2 confidence score 100.0; Supplementary Fig. 4B) and its Harmonin Homology Domain[33] (Phyre2 confidence score 99.8; Supplementary Fig. 4B). These domains in human CCM2 are required to associate with KRIT1/CCM1[34] and MEKK3[28], respectively. Due to their structural homology, we henceforth refer to *Y45F10D.10* as *icap-1* and *K07A9.3* as *ccm-2*.

To determine if these KRI-1 binding partners are required for apoptosis, we knocked down each gene by RNAi in wild type worms, excluding the low abundance binding partner HIP-1, and HSP-17, which might simply be responding to radiation stress. Knockdown of both *ccm-2* and *icap-1* suppressed IR-induced germline apoptosis, while *F37C4.5* did not (Fig. 4c). Since CCM2 and ICAP1 bind separate NPxY/F motifs of KRIT1/CCM1 in vertebrates[34] (Supplementary Fig. 4C), we propose that the *C. elegans* KRI-1 complex contains both CCM-2 and ICAP-1, and inhibits the ERK-5 pathway through MEKK-3, as observed in vertebrates[16,24].

***zipt-2.3* expression is required for IR-induced apoptosis.** To gain more insight into how MPK-2 regulates IR-induced apoptosis, we performed RNA sequencing to identify transcripts altered in *kri-1(ok1251)* animals that return to wild type-like levels of expression in *kri-1(ok1251); mpk-2(ok219)* double mutants. To do this, we treated wild type, *mpk-2(ok219)*, *kri-1 (ok1251)*, and *kri-1(ok1251); mpk-2(ok219)* animals with 60 Gy of IR (Fig. 4d) and identified 784 genes that were significantly increased more than two-fold and 123 genes decreased more than two-fold in *kri-1(ok1251)* worms (Fig. 4e; Supplementary Data 1). To determine if MPK-2 is responsible for the transcriptional changes downstream of *kri-1*, we compared the transcriptomes of *kri-1(ok1251); mpk-2(ok219)* double mutant worms to *kri-1 (ok1251)* animals and found that 629 up-regulated, and 99 down-

regulated transcripts return to wild type-like levels in the double mutants (Fig. 4e; Supplementary Data 2–3). Many of these genes are predicted to be involved in innate immunity and various stress responses (Fig. 4f). To determine which of these mis-regulated genes are required for IR-induced apoptosis, we conducted a targeted RNAi screen using a commercially available RNAi library that covers approximately 85% of the *C. elegans* genome. We systematically knocked down 535/629 genes that were increased two-fold or greater (Supplementary Data 4), and 87/99 genes decreased two-fold or more (Supplementary Data 5). None of the up-regulated genes restored apoptosis when knocked down in *kri-1(ok1251)* mutants, but knockdown of two down-regulated genes in wild type worms suppressed IR-induced germline apoptosis to similar levels as *kri-1* mutants. These genes are *pho-1*, which encodes an intestinal acid phosphatase, and the SLC39 family zinc transporter *zipt-2.3*. We chose to focus our analysis on *zipt-2.3* since high concentrations of zinc can inhibit apoptosis[35,36], and ablation of other zinc transporters in *C. elegans* cause zinc storage defects[37] that result in its diffusion throughout the animal[38]. We confirmed that *zipt-2.3(ok2094)* deletion mutants (Supplementary Fig. 4D) are resistant to IR-induced apoptosis (Supplementary Fig. 4E). To validate the transcriptional changes of *zipt-2.3* detected by RNA sequencing we carried out qPCR to confirm its downregulation in *kri-1 (ok1251)* animals, which increased to wild type levels in *kri-1 (ok1251); mpk-2(ok219)* double mutants (Supplementary Fig. 4F). As expected, *zipt-2.3* levels were restored in *kri-1(ok1251); klf-3 (on34)* double mutants (Supplementary Fig. 4G), indicating that transcriptional regulation of *zipt-2.3* is dependent on KLF-3.

**ZIPT-2.3 in the intestine promotes zinc storage**. To determine the localization of ZIPT-2.3 we created a GFP translational fusion reporter with 5 kb of promoter region of *zipt-2.3*, which expresses ZIPT-2.3::GFP throughout the intestine (Fig. 5a). This expression pattern is consistent with a recent report demonstrating that the *zipt-2.3* promoter drives GFP expression in the intestine[39]. Similar to other intestinal zinc transporters[40] ZIPT-2.3::GFP localizes to gut granules (Fig. 5b), which are the main sites of zinc storage in the animal[37]. To determine if ZIPT-2.3::GFP expression or localization is affected by a loss of *kri-1*, we expressed the array in *kri-1(ok1251)* mutants. Consistent with the RNA sequencing and qPCR results, we observed a strong reduction of ZIPT-2.3::GFP expression with and without IR (Fig. 5c). To determine if ZIPT-2.3 regulates zinc levels or localization, we used the dye Fluozin-3, which detects stored zinc in gut granules[37]. Intriguingly, intestinal zinc signal in gut granules was lost in *kri-1(ok1251)* worms but restored to wild type levels in *kri-1 (ok1251); mpk-2(ok219)*, and *kri-1(ok1251); klf-3(on34)* double mutants (Fig. 5d). As negative controls for gut granule zinc storage, we stained *zipt-2.3(ok2094)* animals as well as *pgp-2(kx48)* mutants that have a reduction of stored zinc due to defective gut granule formation[41]. Collectively, these results indicate that the KRI-1-MPK-2-KLF-3 signaling cascade in the soma ensures proper storage of zinc in the intestine.

***kri-1* mutants have increased zinc in the body cavity**. Next, we wondered if a reduction of stored zinc in the intestines of *kri-1 (ok1251)* mutants results in increased zinc levels throughout the animal. To test this hypothesis, we utilized the dye Zinpyr-1, which specifically detects zinc in the interstitial body cavity of the worm[38]. Consistent with our hypothesis, we detected increased zinc in the body cavity of *kri-1(ok1251)* mutants, compared to wild type, *kri-1(ok1251); mpk-2(ok219)*, and *kri-1(ok1251); klf-3 (on34)* worms that properly store intestinal zinc (Fig. 5d, e). As a positive control for excess body cavity zinc we stained *ttm-1*

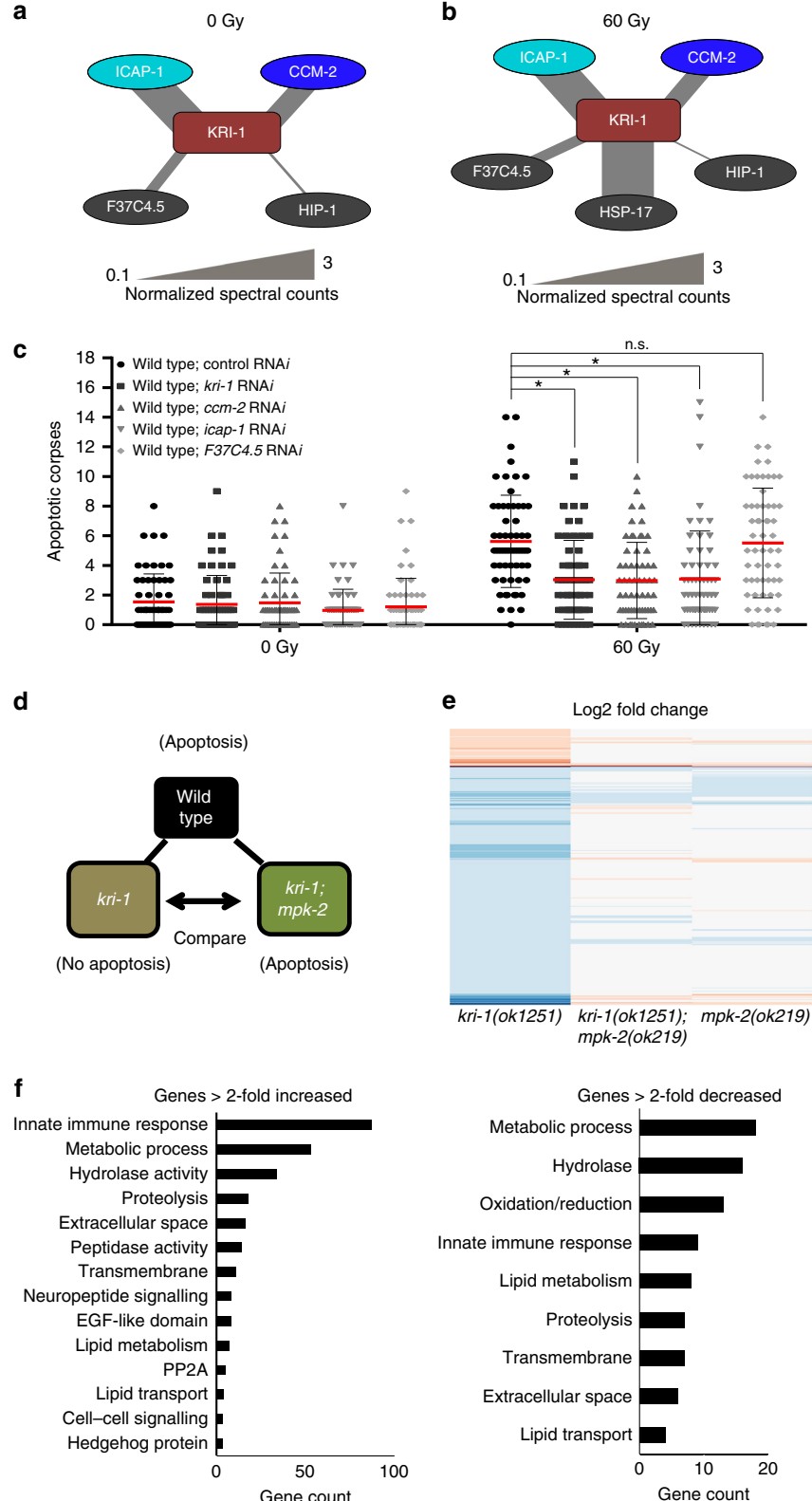

**Fig. 4** KRI-1 interacts with CCM-2, ICAP-1, and regulates transcription. **a**, **b** KRI-1 interacting partners identified by AP-MS. Line thickness represents normalized spectral counts. **c** IR-induced germline apoptosis scored in wild type, wild type; *kri-1* RNA*i*, wild type; *ccm-2* RNA*i*, wild type; *icap-1* RNA*i*, and wild type; *F37C4.5* RNA*i* animals ($n \geq 60$). Red line is mean ± standard deviation. *$P < 0.05$, two-sided, unpaired *t*-test. **d** Schematic of mRNA sequencing. **e** Heat map depicting relative transcript expression in *kri-1(ok1251)*, *mpk-2(ok219)*, and *kri-1(ok1251); mpk-2(ok219)* animals compared to wild type. Red represents down-regulated transcripts, and blue represents up-regulated transcripts. **f** DAVID was used to group transcripts altered in *kri-1(ok1251)* mutants based upon gene ontology categories

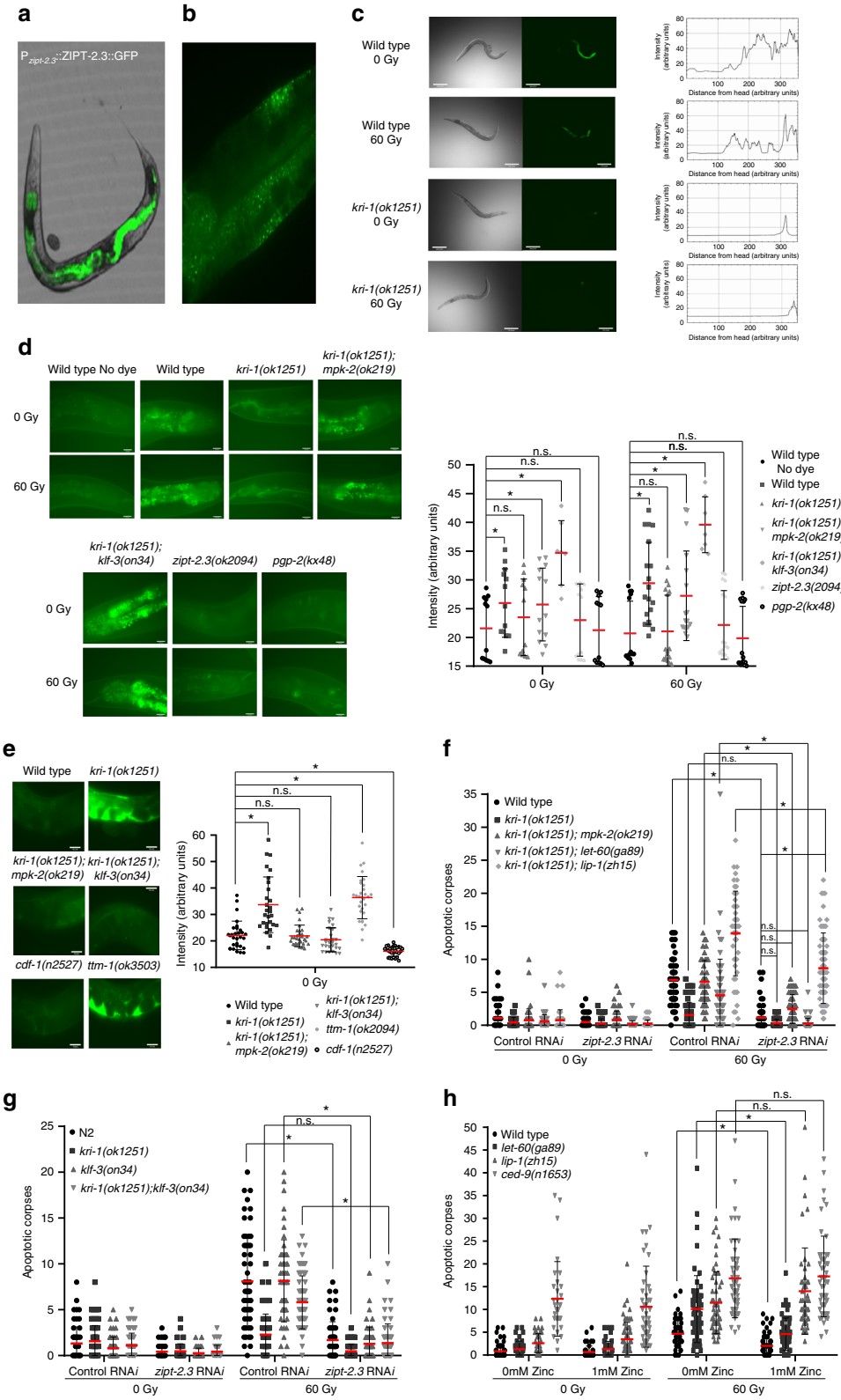

(ok3505) mutants and stained *cdf-1(n2527)* animals for reduced body cavity zinc[38]. These results suggest that the failure to properly store zinc in the intestine of *kri-1(ok1251)* mutants leads to higher levels in the body cavity.

**ZIPT-2.3 promotes IR-induced MPK-1 activation and apoptosis**. To confirm that *zipt-2.3* acts downstream of *mpk-2* and *klf-*

*3*, we knocked-down *zipt-2.3* in *kri-1(ok1251); mpk-2(ok219)*, and *kri-1(ok1251); klf-3(on34)* double mutant worms and observed a similar reduction in apoptosis as knocking-down *zipt-2.3* in wild type animals (Fig. 5f, g). Immunostaining germlines of wild type and *kri-1(ok1251); mpk-2(ok219)* worms for dp-MPK-1 upon *zipt-2.3* RNAi knock-down revealed a reduction of activated MPK-1 (Supplementary Fig. 5A), indicating that *zipt-2.3*

**Fig. 5** ZIPT-2.3 promotes zinc storage and IR-induced apoptosis. **a, b** ZIPT-2.3::GFP expression in wild type animals. Images are representative of three independent lines (**a** is at ×100 magnification and **b** is at ×630 magnification). **c** ZIPT-2.3::GFP in wild type and *kri-1(ok1251)* animals is measured along the intestinal midline from head to tail. Images are representative of three independent replicates ($n \geq 15$). Scale bar is 50 μm. **d** Stored intestinal zinc measured in the intestines of wild type, *kri-1(ok1251)*, *kri-1(ok1251); mpk-2(ok219)*, *kri-1(ok1251); klf-3(on34)*, *zipt-2.3(ok2094)*, and *pgp-2(kx48)* animals ($n \geq 7$). Scale bar is 25 μm. **e** Body cavity zinc measured in wild type, *kri-1(ok1251)*, *kri-1(ok1251); mpk-2(ok219)*, *kri-1(ok1251); klf-3(on34)*, *ttm-1(ok3505)*, and *cdf-1(n2527)* animals ($n \geq 25$). Scale bar is 25 μm. **f** IR-induced germline apoptosis scored in wild type, *kri-1(ok1251)*, *kri-1(ok1251); mpk-2(ok219)*, *kri-1(ok1251); let-60 (ga89)* and *kri-1(ok1251); lip-1(zh15)* animals after knock-down of *zipt-2.3* ($n \geq 55$). **g** IR-induced germline apoptosis scored in wild type, *kri-1(ok1251)*, *klf-3 (on34)*, and *kri-1(ok1251); klf-3(on34)* animals after knock-down of *zipt-2.3* ($n \geq 40$). **h** IR-induced germline apoptosis scored in wild type, *let-60(ga89)*, *lip-1 (zh15)*, and *ced-9(n1653)* animals grown on media supplemented with 1 mM zinc ($n \geq 35$). **d–h** Red line is mean ± standard deviation. *$P < 0.05$, two-sided, unpaired *t*-test

promotes apoptosis via MPK-1 activation. To establish where in the Ras/MAPK pathway *zipt-2.3* functions, we knocked down *zipt-2.3* in *kri-1(ok1251); let-60(ga89)* and *kri-1(ok1251); lip-1 (zh15)* mutants. Reduction of *zipt-2.3* abolished the partial restoration of apoptosis in *kri-1(ok1251); let-60(ga89)* animals but did not suppress the restored activation in *kri-1(ok1251); lip-1 (zh15)* mutants (Fig. 5f). This indicates that ZIPT-2.3 is important for permitting apoptosis downstream of *let-60/Ras*, and upstream of *mpk-1* (Fig. 1c).

**Multiple zinc transporters regulate IR-induced apoptosis.** Given that there are 28 zinc transporter genes in *C. elegans*[42], we reasoned that there are likely zinc transporters in addition to *zipt-2.3* functioning in other tissues, such as the germline, to regulate IR-induced apoptosis. Of the 27 additional transporters, we first investigated *zipt-7.1* since it is expressed in the germline[43] and might be required for transporting zinc in this tissue. However, knockdown of *zipt-7.1* by RNAi did not restore IR-induced apoptosis in *kri-1(ok1251)* mutants, nor was IR-induced apoptosis altered in wild type animals (Supplementary Fig. 5C). Next, we systematically ablated 19/26 of the remaining *C. elegans* zinc transporter genes with available RNAi clones, but none were able to restore apoptosis in *kri-1(ok1251)* mutants (Supplementary Fig. 5B). Interestingly, ablation of *Y105E8A.3*, *zipt-1*, and *zipt-13* suppressed IR-induced apoptosis in wild type animals (Supplementary Fig. 5B). It is possible that these zinc transporters also function to store zinc in the intestine.

**Zinc inhibits IR-induced apoptosis.** Our results thus far are consistent with a model whereby KR1–1 regulates *zipt-2.3* and zinc storage to render the germline competent for apoptosis. To determine if zinc itself affects IR-induced germline apoptosis, we placed juvenile (L4 stage) nematodes on growth media containing a range of exogenous zinc concentrations (0.1, 0.5, and 1 mM) for 24 h. All three concentrations of zinc resulted in an inhibition of IR-induced germline apoptosis (Supplementary Fig. 6A). These animals were otherwise healthy, indicating that the concentrations of zinc tested do not affect other biological processes such as rate of growth or fertility. We also found that 4 h of exposure to 1 mM liquid zinc suppressed IR-induced apoptosis (Supplementary Fig. 6B). The reduction in IR-induced germline apoptosis by exogenous zinc is similar to *kri-1(ok1251)* mutants.

To determine if reducing zinc levels would result in increased IR-induced apoptosis, we incubated worms for 4 h in liquid buffer with the zinc chelator TPEN (N,N,N′,N′-tetrakis (2-pyridinyl-methyl)-1,2-ethanediamine). We chose 25 μM since higher concentrations of TPEN have toxic effects[44]. Exposure to TPEN enhanced IR-induced apoptosis in wild type animals, and restored IR-induced apoptosis in *kri-1(ok1251)* mutants (Supplementary Fig. 6C). To confirm that TPEN reduces body cavity zinc, wild type animals were incubated with the Zinpyr-1 zinc dye and a reduction of fluorescence in the body cavity was observed

(Supplementary Fig. 6D). This indicates that TPEN is effective at chelating zinc, and implicates a link between body cavity zinc and IR-induced germline apoptosis.

Next, we wanted to determine zinc availability in animals by quantitative methods. For this, we used inductively coupled plasma mass spectrometry (ICP-MS)[40] to assess the levels of zinc in populations of wild type, *kri-1(ok1251)*, and *kri-1(ok1251); mpk-2(ok219)* animals. Since we found that the levels of total zinc were the same across all three strains (Supplementary Fig. 6E), defective zinc storage in *kri-1(ok1251)* mutants must result in redistribution to other tissues, as indicated by Fluozin-3 and Zinpyr-1 staining (Fig. 5d, e). To confirm that *kri-1(ok1251)* mutants are unable to store zinc, we incubated animals with 0.1 mM exogenous zinc, and analyzed zinc levels by ICP-MS. As expected, the level of zinc after exogenous exposure was lower in *kri-1(ok1251)* mutants compared to *kri-1(ok1251); mpk-2(ok219)* and wild type animals (Supplementary Fig. S6E). We conclude that *kri-1(ok1251)* mutants are defective in zinc storage, similar to what has been observed with *cdf-2* mutants[38].

**Zinc inhibits IR-induced apoptosis upstream of MPK-1.** To determine if zinc inhibits IR-induced germline apoptosis by suppressing the ERK1/MAPK pathway, we grew *let-60(ga89)/Ras(gf)* and *lip-1(zh15)* animals on 1 mM zinc. Apoptosis was reduced in *let-60(ga89)* but not *lip-1(zh15)* mutants (Fig. 5h), indicating that zinc regulates apoptosis downstream of *let-60/Ras* but upstream of *mpk-1*. Since zinc has been shown to inhibit caspases[36], we wondered whether this was an additional level of regulation in the germline. To test this, we grew *ced-9(n1653)* animals, which have enhanced caspase activation and apoptosis[45] (Supplementary Fig. 5D), on 1 mM zinc, but observed no effects (Fig. 5h). This suggests that in the germline, zinc suppresses apoptosis by inhibiting the MPK-1 MAPK pathway, but not caspase activity. Consistent with this, knockdown of *zipt-2.3* does not reduce apoptosis in *ced-9(n1653)* mutants (Supplementary Fig. 5E).

**Zinc localization and transporters are altered in CCM models.** Given the conservation of the core signaling pathway downstream of KRI-1 we wondered if zinc homeostasis was similarly disrupted in vertebrates. To assess whether vascular zinc levels change in the absence of KRIT1/CCM1, we first determined the ability of Zinpyr-1 to detect zinc in developing wild type zebrafish. By 3 days post fertilization (dpf), zinc was readily detected in the posterior cardinal vein (PCV) and caudal vein (CV) regions of the trunk vasculature, with a similar pattern of localization by 4 dpf (Fig. 6a). By 3 dpf zinc signal is lost in the PCV and CV regions of *krit1/ccm1* mutants (Fig. 6b), suggesting that zinc localization is also regulated by KRIT1/CCM1 in vertebrates.

To investigate the specific contribution of zinc transporter regulation by KRIT1/CCM1, we assessed endothelial-specific conditional *Krit1/Ccm1* null mice using *Pdgfb* promoter driven tamoxifen-inducible Cre recombinase[46], and compared RNA

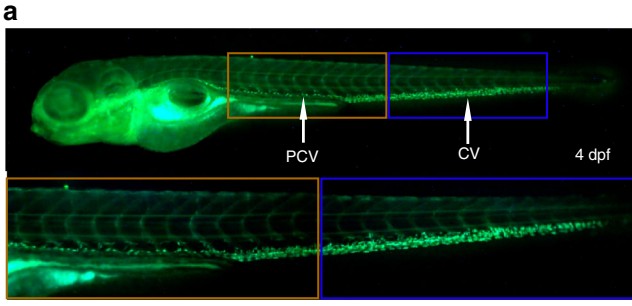

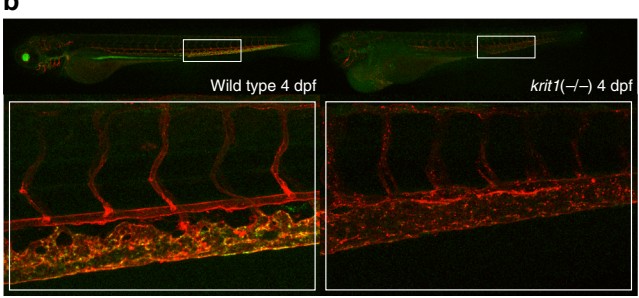

| | Fold change | p-Value | p-Value (FDR adjusted) |
|---|---|---|---|
| **Human** | | | |
| SLC39A10 | −3.27 | <0.0001 | 0.0005 |
| SLC39A11 | 3.44 | 0.002 | 0.03 |
| SLC39A3 | 2.44 | 0.01 | 0.10 |
| **BMEC** | | | |
| **Ccm1** | | | |
| Slc39a4 | −1.56 | 0.0001 | 0.001 |
| Slc39a8 | −1.50 | 0.0003 | 0.004 |
| Slc39a3 | 1.14 | 0.02 | 0.11 |

**Fig. 6** Zinc localization and *SLC39* zinc transporter expression in CCM models. **a** Zinypr-1 detects zinc in the posterior cardinal vein (PCV) and caudal vein (CV) vasculature of wild type zebrafish 4 days post fertilization (dpf). **b** Zinc localization in the PCV and CV is lost in the absence of *krit1/ccm1* 4 dpf. The endothelial cell vasculature is marked in red, and zinc appears green (yellow overlay). **c** *SLC39* zinc transporter expression in CCM lesion tissue from patients (human) and in brain microvasculature endothelial tissue (BMEC) of endothelial-specific knockout mice (BMEC Ccm1)

## Discussion

In this study, we set out to determine how the KRI-1 scaffold protein transmits pro-apoptosis signals from the soma (intestine) to the germline in *C. elegans*. By taking an unbiased forward

genetic approach we uncovered three genes that comprise a canonical ERK-5/MAPK pathway, as well as the KLF-3 transcription factor, that functions in the intestine with KRI-1 to regulate apoptosis (Fig. 7). In vertebrates, the ERK5/MAPK pathway has been shown to be suppressed by the KRI-1 homologue KRIT1/CCM1 in conjunction with its binding partner CCM2. The CCM2 C-terminal Harmonin domain interacts with MEKK3[28], and ERK5 is hyperactivated downstream of MEKK3 when KRIT1 is ablated in the zebrafish myocardium[16], as well as the mouse endothelium[24]. This over-activation of ERK5 in the mouse endothelium results in the formation of CCM lesions. Given that auto-phosphorylation of MEKK3, and subsequent activation of ERK5 is increased in the absence of CCM2[47], it is likely that in *C. elegans* KRI-1 functions to attenuate ERK-5 signaling by a similar mechanism through MEKK-3. Our identification of a *C. elegans* CCM2 ortholog, along with the KRIT1 binding partner ortholog Y45F10D.10/ICAP1, demonstrates that the KRIT1/CCM1 signaling complex is conserved across species. In mice, endothelial-specific ablation of *ERK5* results in apoptotic cell death[48] and in HeLa cells ERK5 suppresses IR-induced apoptosis[49]. Therefore, it is possible that the anti-apoptotic function of ERK5 signaling might contribute to the pathology of CCM disease.

In vertebrates, over-activation of ERK5 results in increased transcript levels of *KLF2* and *KLF4*[15,50], which is predicted to contribute to CCM disease[24]. While *C. elegans klf-3* transcript levels were not altered in *kri-1(ok1251)* animals (Supplementary Data 1), we identified a gain-of-function mutation in KLF-3 that restores IR-induced apoptosis in *kri-1(ok1251)* mutants. Despite the potential differences between worms and vertebrates in how ERK5 regulates KLFs, we hypothesize that the gain-of-function mutation in KLF-3 expressed in the intestine[51] restores normal transcription in the absence of KRI-1. We discovered many genes that were transcriptionally altered in *kri-1(ok1251)* mutants, including genes affecting non-apoptotic processes such as innate immunity, suggesting that KRI-1-ERK-5-KLF-3 signaling has many other roles in the cell. For example, KRI-1 has been shown to regulate lifespan by a mechanism that is independent of apoptosis[12,14]. Thus, in the future it will be interesting to determine if the other biological functions of KRI-1 require these altered genes. Analysis of differentially regulated transcripts in *kri-1(ok1251)* mutants uncovered the *SLC39* zinc transporter *zipt-2.3*, which we confirmed to be regulated by KLF-3. The established links between zinc transporters and apoptosis[52], and the possibility that zinc acts cell non-autonomously as a signaling molecule prompted us to examine the role of zinc in germline apoptosis.

Recently, *zipt-2.3* was shown to be transcriptionally regulated in response to zinc stress[39]. Here, we observed a striking decrease of stored zinc in vesicular gut granules of *kri-1(ok1251)* as well as *zipt-2.3(ok2094)* mutants (Fig. 5d), demonstrating that ZIPT-2.3 is dynamically regulated by KRI-1 to promote zinc storage. Zinc is required for the normal functions of approximately 3000 proteins, reinforcing the importance of this cation in cellular and developmental biology[53]. This is apparent from increased levels of zinc resulting in lifespan, developmental, reproductive, locomotive, and chemotaxis defects[54] in *C. elegans*. Furthermore, zinc exposure activates stress responses in most tissues, and behavioral toxicities were found to be transferred to progeny[54]. Thus, *C. elegans* employs strict homeostatic mechanisms to prevent excess zinc from inducing adverse effects, such as inhibiting stress-induced apoptosis. For instance, the histidine ammonia lyase, HALY-1, regulates zinc tolerance by modulating levels of histidine[55]. Since histidine functions as a zinc chelator, higher levels of this amino acid can reduce the toxic effects of excess zinc[55]. Inversely, reducing zinc levels in *C. elegans* with zinc chelators

from isolated brain microvascular endothelial cells (BMEC) with wild-type mice. In *Krit1/Ccm1* mouse mutant BMECs there was a significant downregulation of *Slc39a4* and *Slc39a8* transcripts (Fig. 6c), and a slight upregulation of *Slc39a3*. To determine if the transcriptional regulation of *SLC39* zinc transporters is relevant to the pathology of CCM disease, we analyzed endothelial cell (EC) RNA from five human CCM lesions (four sporadic and one familial CCM1) and compared these samples to corresponding brain tissue from three patients free of neurological disease. Transcriptome analyses showed that *SLC39A10* was downregulated in ECs from human lesions compared to normal ECs from capillaries (Fig. 6c). In addition, *SLC39A11* and *SLCA39A3* were up-regulated in human CCM lesions (Fig. 6c). While this demonstrates that *SLC39* zinc transporter genes are mis-regulated in the absence of CCM1 in *C. elegans*, mice, and humans, the extent to which this might contribute to CCM disease remains to be investigated.

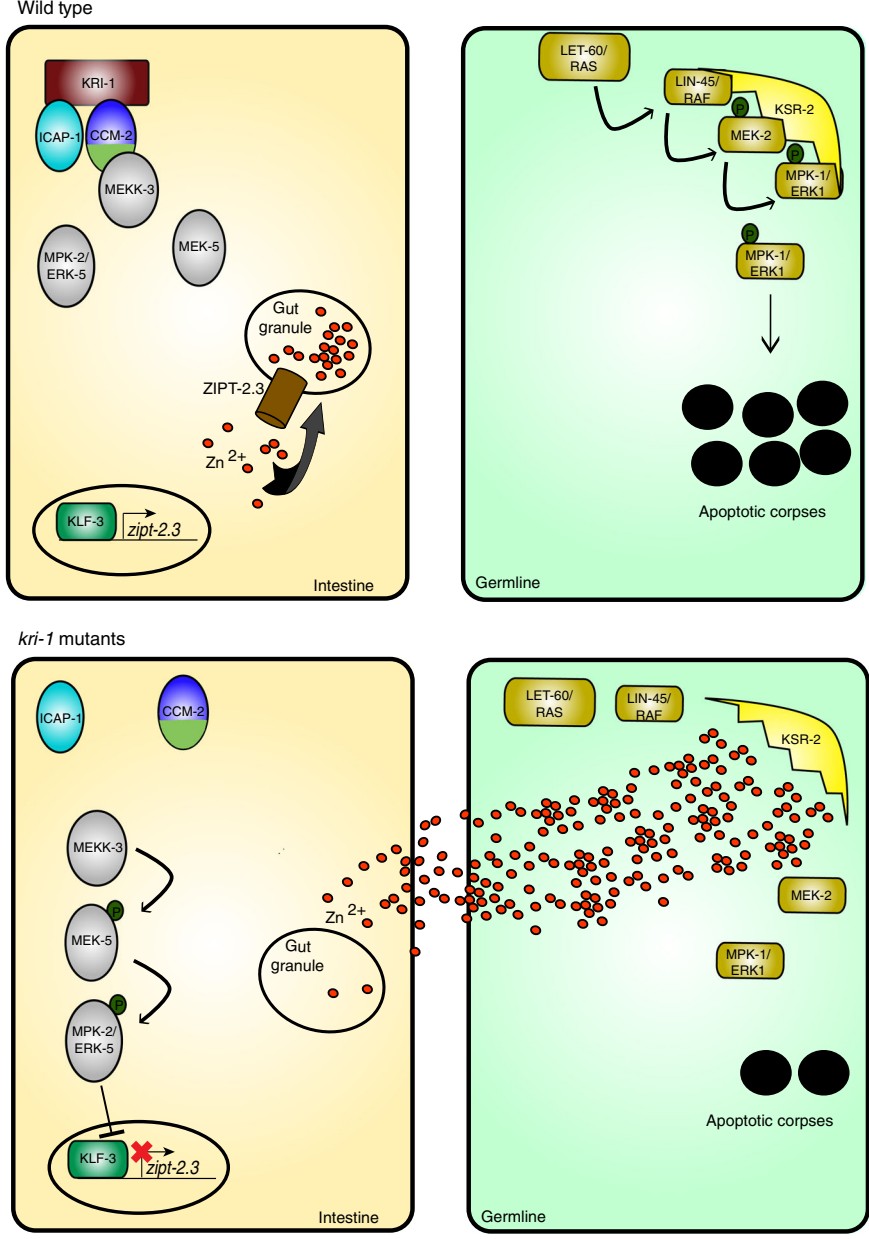

**Fig. 7** KRI-1 promotes IR-induced apoptosis by regulating zinc. Schematic depicting the mechanism by which intestinal KRI-1 regulates germline MPK-1 and apoptosis. KRI-1 in complex with CCM-2 prevents the over-activation of MPK-2, allowing the transcription of *zipt-2.3* by KLF-3. ZIPT-2.3 ensures the storage of zinc in the intestinal gut granules, which permits MPK-1 activation to promote apoptosis

results in increased lifespan, which is correlated with improved proteostasis[56]. Collectively, this indicates that finely balanced zinc levels are critical to cellular health, which we also found to be true for apoptosis regulation. In vertebrates, zinc has been shown to prevent apoptosis by inhibiting the function of pro-apoptotic multi-BH3 domain proteins such as BAX and BAK[35], and Caspase-3[36]. The absence of BAX and BAK proteins in *C. elegans*[57] and the inability of *zipt-2.3* knockdown or high zinc levels to suppress apoptosis in *ced-9(n1653)* animals (Fig. 5h; Supplementary Fig. 5D) indicates that zinc does not likely affect the core apoptosis pathway in this animal.

In the context of *C. elegans* vulval development, excess zinc and loss of zinc transporters have been shown to suppress the ERK1/MAPK pathway downstream of LET-60/RAS[58], and upstream of MPK-1[59]. Consistent with these observations, we found that regulation of *zipt-2.3* and zinc affects IR-induced germline

apoptosis downstream of *let-60*, and upstream of *mpk-1*. We detected increased zinc throughout the body in *kri-1(ok1251)* animals and confirmed by ICP-MS that *kri-1(ok1251)* animals are unable to store zinc. Furthermore, we were able to restore IR-induced apoptosis in *kri-1(ok1251)* mutants by chelating non-stored zinc with TPEN. It is possible that failure to store zinc in intestinal vesicles results in leakage into the germline, where it could inhibit ERK1/MAPK signaling (Fig. 7). Since zinc has been shown to inhibit the function of RAF[60] and KSR[59] in *C. elegans*, it is possible that zinc suppresses MPK-1 activation in the germline at either of these steps in the pathway.

While we show that the KRIT1/CCM1 cascade is conserved from nematodes to humans, it remains to be determined whether zinc regulation is relevant to CCM disease in humans. Encouragingly, loss of zinc in the PCV and CV vasculature in *krit1/ccm1* mutant zebrafish suggests a conserved role for KRIT1 in zinc

regulation in vertebrates. In zebrafish, zinc is known to affect brain acetylcholine levels, memory and motor activities[61], while zinc chelation is neuroprotective in a zebrafish model of hypoxic brain injury[62]. These findings reveal that maintaining proper zinc levels might be important for normal brain function, and possibly to prevent the development of neurological diseases such as CCM.

Surveying zinc transporter gene regulation in *Krit1/Ccm1* mouse BMECs and patient CCM tissue revealed mis-expression of several transporters, consistent with our observations in *C. elegans*. Intriguingly, a recent report described a role for endothelial-expressed SLC39A8 in regulating mouse myocardium development by promoting proper zinc localization[63]. Since myocardium development is altered in a similar manner upon loss of CCM proteins in zebrafish[15,16], and given that we found *Slc39a8* to be down-regulated in *Krit1/Ccm1* mouse BMECs (Fig. 6c), it is possible that aberrant zinc regulation plays a role in the development of CCM lesions. Future studies will be important to test this experimentally using zinc transporter mutant mouse models.

In summary, we have uncovered KRI-1 as being part of a conserved CCM complex that promotes germline apoptosis non-autonomously from the intestine. Additionally, since this core complex is conserved across species, we suggest that the discovery of KRI-1/KRIT1 linked zinc regulation, through zinc transporters, may impact the pathology of CCM disease.

## Methods

**Contact for reagent and resource sharing**. Any queries and reagent requests should be sent to the corresponding author.

**Experimental model and subject details**. All nematode strains were maintained at 16 °C and kept at 20 °C for experimentation on NGM (nematode growth medium) agar plates with OP50 *Escherichia coli* as a food source, following standard procedures. N2 Bristol was used as the wild type strain and mutant strains are listed in Supplementary Table 1. HT115 *E. coli* was used for RNAi knockdown. Zebrafish were maintained under standard conditions and strains used in this study are listed in Supplementary Table 1. Zebrafish embryos or larvae were treated with 20 μM Zinpyr-1 for 30 min, prior to imaging. The embryos were treated with 1-phenyl-2-thiourea (PTU) prior to the appearance of pigmentation. Handling of zebrafish was performed in compliance with German and Brandenburg state law, carefully monitored by the local authority for animal protection (LUVG, Brandenburg, Germany; Animal Protocol #2347-18-2015).

**RNAi feeding in *C. elegans***. The RNAi clones in this study were obtained from Source BioSciences, which covers approximately 85% of the *C. elegans* genome. Bacteria colonies were grown overnight in LB with 100 μg/mL ampicillin and 10 μg/mL tetracycline at 37 °C and concentrated 10× the next day. NGM plates were seeded with 100 μl of the concentrated culture and synchronous L1 animals were grown at 20 °C. As a control HT115 containing the L4440 vector with the non-expressed gene *Y95B8A_84.g* was used.

**Germline apoptosis quantification**. Worms were irradiated 24 h post the L4 stage using a C[137] source and apoptotic germ cells quantified 24 h post irradiation (60 Gy). Animals were mounted on 4% agarose pads on glass slides with 20 mM L-tetramisole in M9 buffer. Apoptotic cells were scored in one germline arm by manually counting the number of raised and refractile corpses using a Leica DMRA2 system (Wetzlar, Germany) utilizing standard Differential Interference Contrast optics.

**Statistical analyses**. Microsoft Excel was used to determine statistical significance using a two-sided Student's *t*-test, assuming equal variance. Data was considered significant when the *P*-value was less than 0.05. The mean (red bar) ± standard deviation was included on all dot plots. All experiments were repeated three times, unless otherwise stated.

**Imaging**. All imaging with *C. elegans* was carried out with a Leica DMRA2 system (Wetzlar, Germany) using standard differential interference contrast optics and fluorescent channels. Fluorescence intensity was measured using ImageJ software (version 1.52a). Zebrafish embryos were imaged with a LSM 710 confocal microscope (Zeiss).

**Germline dissection and p-MPK-1 staining**. Young adult worms (24 h post L4) were irradiated with either 60 or 0 Gy of IR, and washed once in 1× PBS 3 h post IR, and transferred to 30 μl 1× PBS + 4 mM Tetramisol on slides coated with 20 μl of 0.1% w/v polylysine. Germlines were dissected by removing the heads of animals, using two 27 G needles. 1× PBS + 4 mM Tetramisol was replaced with 30 μl fresh 2% paraformaldehyde in 1× PBS for 10 min at room temperature. Paraformaldehyde was prepared by dissolving 0.04 g dry paraformaldehyde in 1 mL dH$_2$0 + 2μl 1 M NaOH at 65 °C for 1 h with occasional mixing, followed by removing 500 μl from the tube, and replacing with 500 μl of H$_2$PO$_4$ buffer (0.004 g KH$_2$PO$_4$, 0.0188 g NaHPO$_4$, 500 μl H$_2$O, pH = 7.2). After germlines were isolated, coverslips were added to slides and placed on dry ice for at least 5 min, followed by freeze-cracking by rapidly removing the coverslip. Germlines were then immediately fixed by immersing the slides in 100% methanol for 5 min and transferred to a 1:1 mixture of methanol:acetone for 5 min. Finally, to end fixation, slides were transferred to 100% acetone for 5 min. Slides were then washed with 30 μl 1× PBS + 0.1% Tween20 "PBST" for 10 min, three times and incubated with one drop of Image-it added to each slide, for 20 min. Germlines were blocked with 30 μl PBST + 1% BSA for 1 h, followed by overnight incubation with 30 μl 1:100 anti-dp-Erk-1 plus 1:100 anti-Nuclear Pore Complex in (PBST + 1% BSA) (Supplementary Table 2) at room temperature with parafilm over the slide. The next day, parafilm was removed, and slides washed three times with 30 μl 1× PBST for 10 min. 30 μl of secondary 1:500 goat anti-rabbit Alexa 488 for p-Erk1, plus 1:500 donkey anti-mouse Alexa 568 for Mab414 in (PBST + 1% BSA) were added to each slide, incubating at room temperature for 1 h. Slides were then washed once with 30 μl 1× PBST for 10 min and incubated with 30 μl DAPI for 10 min (1:1000 of 1 mg/mL DAPI in PBST) followed by a final wash with 30 μl 1× PBST for 10 min. PBST was removed and 5 μl of Prolong Gold was added to each slide with a coverslip and sealed.

**EMS mutagenesis and F2 starvation screen**. *kri-1(ok1251)* late L4/young adult worms were incubated with 50 mM EMS (Supplementary Table 3) for 4 h in 15 mL conical tubes and allowed to lay eggs on NGM plates. Approximately 500,000 F1 worms (1,000,000 haploid genomes) were divided into 20 separate populations and left to lay F2 eggs. These eggs were transferred to twenty 50 mL conical tubes with liquid M9 buffer (one tube for each population) and allowed to hatch into L1 larvae. If left under these conditions without food, the larvae arrest growth, and remain in a state of diapause. *kri-1(ok1251)* F2 larvae were kept in M9 for 1 week, and the entire population from each of the 20 tubes was transferred onto 20 large plates with growth media (one plate for each tube). Since *kri-1(ok1251)* L1 larvae normally do not survive a week of diapause, only animals with mutations that suppress this phenotype grew to adulthood. These survivors were singled out and clonal lines were established by self-fertilization. F3 progeny from these single survivors were assessed for a restoration of IR-induced apoptosis compared to non-irradiated controls. A single suppressor line was chosen from any given starting population, such that 13 candidates from the 20 populations were sent for whole genome sequencing.

**Whole genome sequencing and analysis**. DNA was isolated using the DNeasy Blood & Tissue Kit (Supplementary Table 4) and barcoded libraries were created for each strain using the Nextera DNA Sample Preparation Kit from Illumina, and a HiSeq2500 instrument was used to produce 75 base paired-end reads. First, the sequencing reads were examined for sequence quality using FastQC (http://www.bioinformatics.babraham.ac.uk/projects/fastqc/) to ensure good quality up to the full 75 bases. Bases lower than a quality threshold of 30 were trimmed off using Trimmomatic, however very little trimming was required. Reads were aligned to the *C. elegans* N2 reference genome (release W220) using BWA-mem. Alignments were sorted by coordinate order and duplicates removed using Picard (http://picard.sourceforge.net). Before variant calling, reads were processed in Genome Analysis Tool Kit (GATK) v2.5 for indel realignment and base quality score recalibration, using known *C. elegans* variants from dbSNP build 138 (http://www.ncbi.nlm.nih.gov/SNP/). GATK HaplotypeCaller was used to call variants, and results were filtered for a Phred-scaled *Q* score > 30. For each SNP a call is made based on (1) the number of reads, (2) the Phred score of the reads, and (3) how many of each alternate allele is identified. For each SNP an entry was made giving the zygosity-quality(*Q*)score-#reads. Finally, called variants were annotated using Annovar to obtain a list of variants for each sample.

**Cloning**. To generate the P*mpk-2*::*gfp* transcriptional reporter 6 kb upstream of the *mpk-2* start codon was amplified from wild type genomic DNA and ligated into the SphI and AgeI restriction sites of the Fire Kit vector pPD95.75. The P*mpk-2*::*mpk-2* construct was generated by replacing gfp in the above P*mpk-2*::*gfp* plasmid with *mpk-2* amplified from wild type genomic DNA and ligated into the AgeI and ApaI restriction sites of the above plasmid. The correct sequence of *mpk-2* was verified by Sanger sequencing. The P*elt-2*::*mpk-2* construct was generated by ligating *mpk-2* into the EagI and ApaI restriction sites of pJM559 which contains 5 kb of the elt-2 promoter. The P*zipt-2.3*::*zipt-2.3*::*gfp* translational fusion construct was generated by amplifying P*zipt-2.3*::*zipt-2.3* starting from 5 kb upstream of the *zipt-2.3* start codon and ending after the 3′UTR of *zipt-2.3* from wild type genomic DNA. This product was then inserted into BamHI and KpnI restriction sites of the Fire Kit

vector pPD95.75. The correct sequence of *zipt-2.3* was verified by Sanger sequencing. Cloning was conducted using NEB restriction enzymes and buffers, and Invitrogen T4 ligase and buffer. All primers used are listed in Supplementary Table 5.

**Generation of transgenic strains**. Constructs were initially injected into wild type N2 worms at a concentration of 50 ng/µl to create extra-chromosomal arrays. The plasmids pCFJ90 (P*myo-2::mcherry*) and pCFJ104 (P*myo-3::mcherry*) were used as co-injection markers for P*mpk-2::mpk-2* and P*elt-2::mpk-2* and injected at concentrations of 2.5 and 5 ng/µl, respectively.

**RNA isolation**. 2000 Synchronized L1 worms per condition were grown in biological triplicate and irradiated 24 h post L4 and transferred to 1.5 mL conical tubes using M9 buffer 3–6 h post 60 Gy of IR. Worms were washed three times with M9 buffer, M9 removed, and 1 mL of Trizol added. Worms in Trizol were flash frozen using liquid nitrogen and subjected to three rounds of freeze-thawing. 200 µl of chloroform was added, and samples and vortexed for 30 s. Samples were then centrifuged at 12,000×*g* for 15 min at 4 °C. The aqueous phase was transferred to new tubes, and 500 µl of 100% cold isopropanol added to samples and left to precipitate overnight at −20 °C. The next day, samples were centrifuged at 12,000×*g* for 15 min at 4 °C, and supernatant was removed from the tubes. RNA pellets were washed with 1 mL of 75% ethanol in DEPC dH₂O and vortexed for 1 min. Samples were centrifuged at 12,000×*g* for 5 min at 4 °C and supernatant discarded. The RNA pellet was air dried for 5 min and resuspended in 50 µl of Invitrogen Ultrapure dH₂0.

**RNA sequencing and analysis**. mRNA was purified from total RNA by the Hospital for Sick Children Next Generation Sequencing Facility (TCAG) and reads were compared to the wild type samples using Tophat/Cufflinks. Sequencing reads were analyzed for quality with Fastqc. Trimmomatic was used to eliminate low-quality reads and remove adapters. Specific to *C. elegans*, the minimum intron length in Tophat was lowered to 30 bp from the default 70 due to small intron sizes in *C. elegans*. The reference genome and transcript annotations were based on release WS235 (CE11).

**Quantitative RT-PCR**. To determine changes in *zipt-2.3* mRNA levels in *C. elegans*, total RNA was extracted as described above, 3–6 h post 0 Gy or 60 Gy. cDNA was prepared from 500 ng of total RNA using the Invitrogen SuperScriptIII First-Strand Synthesis System with total RNA primed with Random Primer Mix. The reaction was incubated at 25 °C for 10 min, 50 °C for 60 min, and 85 °C for 5 min. 1 U of RNaseH was added per sample and incubated at 37 °C for 30 min. *zipt-2.3* transcript levels were quantified in wild type, *kri-1(ok1251), kri-1(ok1251); mpk-2 (ok219), mpk-2(ok219), kri-1; klf-3(on34),* and *klf-3(on34)* animals using the BioRad SYBR Green Supermix and the BioRad CFX96 Real-Time PCR Detection System. Transcript levels were normalized to an internal tubulin control, and repeated with two biological replicates. The cDNA was amplified by qPCR using the following conditions: 95 °C for 30 s, 95 °C for 15 s, 60 °C, and repeated for 40 cycles.

**Zinc staining with Fluozin-3**. 10 µM Fluozin-3 was added to NGM plates with OP50 *E. coli* and left to dry in the dark for 30 min. L4 stage worms were added to the plates and incubated in the dark at 20 °C for 24 h. Worms were transferred to new NGM plates without dye for 30 min to de-stain, either irradiated (60 Gy) or left unirradiated, and imaged 3–6 h post IR. To distinguish Fluozin-3 fluorescence from gut granule autofluorescence, wild type worms without dye were imaged.

**Zinc staining with Zinpyr-1**. 100 µM ZnSO₄ was added to NGM plates with OP50 *E. coli*. L4 stage worms were added to the plates and left at 20 °C for 20 h. 20 µM Zinpyr-1 was added to the plates, and worms incubated in the dark at 20 °C for 2 h. Worms were transferred to new NGM plates without dye for 30 min to de-stain and immediately imaged.

**ICP-MS**. 5000 Synchronized L1 worms were grown in biological triplicate on NGM and NGM + 0.1 mM ZnSO₄ plates. Young adult worms were collected by washing three times in 10 mL M9 + 0.1% Triton X-100. Worms were pelleted in 100 µl M9 buffer and stored at −80 °C. Worm pellets were then freeze-thawed three times using dry ice with 100% ethanol and 500 µl of Droso buffer + 0.1 mM DTT + 0.1% NP40. Worm pellets were sonicated in a water bath for 30 cycles (30 s on, followed by 30 s off). 10 µl of the worm lysate was used to determine protein concentration using the Bradford assay. 300 µl of trace metal grade nitric acid and 100 µl of trace metal grade hydrogen peroxide were added to 600 µl of worm lysate. Worm lysates were sonicated again for 20 cycles. Finally, the samples were diluted to 2% nitric acid in a final 15 mL volume by adding each sample to 14 mL ultra pure dH₂O. Zinc standards were also prepared. Zinc levels were measured by ICP-MS at the University of Toronto (Mississauga campus) Core Instrument Facility.

**CRISPR/Cas9 genome editing**. Wild type young adult hermaphrodites were injected with crRNA and repair templates for *klf-3(on34)* and *dpy-10* as a marker

for positive selection. Dumpy and Roller F1 worms were singled out onto individual plates and screened for the *klf-3(on34)* mutation by PCR and sequencing. The positive candidate was outcrossed two times to remove the *dpy-10* mutation. Oligonucleotides used are listed in Supplementary Table 5.

**Affinity purification and mass spectrometry**. 800,000 wild type, KRI-1::GFP, and ACT-5::GFP expressing strains were each grown in biological triplicate on ten large 90 mm NGM plates with OP50 *E. coli*. Young adult worms were either irradiated (60 Gy) or left unirradiated. Worms were collected 6 h later by washing with M9 buffer and transferred to a 15 mL conical tube followed by two more M9 washes. To remove bacteria from the intestines, worms were incubated and rocked for 30 min in 10 mL M9 buffer. Afterwards, worms were washed three times more in M9 followed by two washes in autoclaved dH₂O and pelleted. Worm pellets were flash-frozen in liquid nitrogen and stored at −80 °C.

To begin GFP affinity purification, worm pellets were thawed on ice in ice-cold DROSO lysis buffer (30 mM HEPES pH 7.4, 100 mM Potassium Acetate, 2 mM Magnesium Acetate, 0.1% NP-0.4, 2 mM DTT), with one Mini Protease Inhibitor Cocktail tablet, 1:100 Phosphatase Inhibitor 2, 1:100 Phosphatase Inhibitor 3. Worms were lysed by a dounce homogenization in a chilled Wheaton Dounce Homogenizer (metal) and the lysates were centrifuged at 10,000×*g* for 30 min at 4 °C, and the supernatant transferred to fresh tubes. For affinity purification, 10 mg of protein from each sample was incubated for 3 h at 4 °C on a nutator in 30 µl of a 50% slurry of GFP-Trap®_MA beads pre-washed in DROSO buffer. Using a magnetic rack to secure the beads, the supernatant was removed, and the beads were then transferred to a new 1.5 mL microcentrifuge tube in DROSO buffer. Beads were washed once in TAP buffer (50 mM HEPES-KOH pH 8.0, 100 mM KCl, 2 mM EDTA, 0.1% NP-0.4, and 10% glycerol), and once in 50 mM ammonium bicarbonate, pH 8. Supernatant was removed and beads were resuspended in 10 µl of 50 mM ammonium bicarbonate containing 1 µg of mass spectrometry-grade trypsin. Samples were digested on beads overnight at 37 °C on a rotator. The next day, each sample was spun down at 400×*g* for 1 min to pellet beads and the supernatant transferred to a new 1.5 mL microcentrifuge tube. This supernatant was further digested for 4 h by supplanting with 0.25 µg of trypsin. The digestion was stopped by adding 1 µl of 50% formic acid, and samples were lyophilized and stored at −80 °C. Samples were then sent for mass spectrometry at the Lunenfeld-Tanenbaum Research Institute in Toronto, Canada.

All proteins were analyzed with Significance Analysis of INTeractome (SAINT) using SAINTexpress (version 3.6.1) to identify true interaction partners using default parameters. Proteins with a probability ≥ 0.95 were considered, corresponding to an estimated protein level false-discovery rate (FDR) of approximately 0.5%.

**Sequence and structure analysis**. The sequences of the KRI-1 binding partners identified by mass spectrometry were analyzed by unsupervised threading using Phyre2 (Protein Homology/analogY Recognition Engine v2.0; http://www.sbg.bio.ic.ac.uk/phyre2/html/page.cgi?id=index) using the "intensive" model with default options. For each sequence/structure displayed, the model with the highest prediction for each domain was selected, and a complete structural prediction was generated by Phyre2. Starting from K07A9.3, the highest matching template was PDB:4WJ7 "CCM2 PTB domain in complex with KRIT1 NPxY/F3", with a confidence score of 100.0 despite a modest 20% sequence identity. The second highest matching template was PDB:4FQN "Crystal structure of the CCM2 C-terminal Harmonin Homology Domain (HHD)", with a confidence score of 99.8, and 21% sequence identity. For Y45F10D.10, the top hit was PDB 4DX9: "ICAP1 in complex with integrin beta 1 cytoplasmic tail", with a confidence score of 100.0 and 25% sequence identity. Sequence alignments to these structural templates were generated in Phyre2, and manually converted to FASTA ALN for image generation in BoxShade (https://embnet.vital-it.ch/software/BOX_form.html). Structural visualization was in PyMOL (PyMOL Molecular Graphics Systems v 2.0.7).

**Transcriptomes of human CCM lesions**. Five human CCM lesions were surgically resected for clinical indications unrelated to the research. The specimens were snap frozen in the operating room upon excision, embedded in optimal cutting temperature (OCT), and stored at −80 °C. Brain tissue from 3 patients free of neurological disease was acquired during autopsy, fixed in formalin and embedded in paraffin blocks (FFPE). Sections of 5 µM of the tissue samples were processed, mounted on Leica glass slides (Leica Biosystems, Wetzlar, Germany), and stained for frozen tissue. The endothelial cells (ECs) of CCMs and normal capillaries were then collected using laser capture microdissection (LCM; Leica LMD 6500) at 40× of magnification, and then stored at −80 °C. The RNAs of the collected endothelial cells were extracted using TRIzol. RNAs from human lesion and control samples were sequenced in 2 batches with 47 basepair (bp) single-end reads. RNA libraries were generated, multiplexed and sequenced on Illumina HiSeq4000 platform. The quality of raw sequencing reads was then assessed by FastQC (available at http://www.bioinformatics.babraham.ac.uk/projects/fastqc/). The post-alignment quality control was evaluated using RSeQC and Picard tools. Finally, reads were mapped to UCSC human genome model (hg19) using STAR. Gene transcripts were assembled and quantified based on the corresponding human genome (UCSC RefSeq hg19) using count-based method featureCounts. Differentially expressed genes (DEGs)

analysis were conducted using DESeq2, with an additive model for batch effect correction when needed. The DEGs were identified at a standard statistically significant threshold of false discovery adjusted P-value (FDR) of 0.05 and fold change of 1.2. Human sample collection complied with all relevant ethical regulations. The study protocol was approved by the Institutional Review Board at University of Chicago Medicine, and informed consent was obtained from each subject.

**Mouse brain microvascular endothelial cell transcriptomes**. Three endothelial-specific conditional *Krit1* null mice were generated using a *Pdgfb* promoter driven tamoxifen-inducible Cre recombinase, *Pdgfb-iCreERT2*; *Krit1fl/fl*. Three other wild-type mice were used as controls. BMECs were isolated and maintained at 37 °C in 5% $CO_2$ and 95% air and treated with 5 μM of 4-hydroxytamoxifen for 48 h to induce allelic loss in BMECs harboring the Cre-recombinase system and floxed alleles. RNAs were extracted using the TRIzol protocol. Illumina's TruSeq Stranded mRNA Sample Prep Kit was used for libraries generation. RNA samples of *Krit1* treated mice together with their corresponding null mice were sequenced with 100 bp single-end sequencing reads. The quality of raw sequencing reads was assessed using FastQC v0.11.2 and the post-alignment QC was evaluated with RSeQC and Picard tools v1.117 (http://broadinstitute.github.io/picard/). Reads were mapped to Gencode mouse genome model (GRCm38 V13) using STAR STAR_2.5.3a. Gene transcripts were assembled and quantified on the corresponding mouse genome annotation file using count-based method featureCounts. DEGs analysis were conducted with DESeq2, using an additive model for batch effect correction when needed. DEGs were identified at a standard statistically significant threshold of false discovery corrected P-value (FDR) of 0.05 and fold change of 1.2. Gene Ontology enrichment analysis was conducted with R bioconductor package clusterProfiler based on the GO database on mouse species.

**Reporting summary**. Further information on experimental design is available in the Nature Research Reporting Summary linked to this article.

## Data availability

All raw mass spectrometry data and SAINTexpress result tables are deposited in the MassIVE repository housed at the Center for Computational Mass Spectrometry at UCSD (https://massive.ucsd.edu/ProteoSAFe/static/massive.jsp). The dataset has been assigned the accession number MSV000082400. The dataset was also contributed to the ProteomeXchange Consortium with identifier PXD009886.

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

## Acknowledgements

The authors are grateful to Dr. Peter Roy for advice on the project, Dr. Abigail Mateo for helping to design Fig. 7, and members of the Derry lab for comments on the manuscript. This work was supported by an operating grant from the Canadian Institutes of Health Research (CIHR) to W.B.D. (MOP 137089). A.-C.G. holds a Tier 1 Canada Research Chair in "Functional Proteomics" and was supported by a CIHR Foundation grant (FDN 143301). I.A.A. was supported by an operating grant from the National Institutes of Health (P01 NS092521). S.A.-S. was supported by the Excellence cluster REBIRTH (SFB958) by Deutsche Forschungsgemeinschaft (DFG) projects SE2016/7-2 and SE2016/10-1 and by the DZHK. E.M.C. was supported by an Alexander Graham Bell Canada Graduate Scholarship from the Natural Sciences and Engineering Council of Canada and a Terry Fox Foundation Strategic Training in Transdisciplinary Radiation Science Scholarship. R.G. was supported by a Safadi Translational Fellowship and J.K. was supported by the Sigrid Juselius Foundation.

## Author contributions

E.M.C. designed the experiments, performed experiments, analyzed the data, and wrote the manuscript. B.L. performed apoptosis experiments. Y.O. performed cloning and apoptosis experiments and B.Y. performed CRISPR gene editing and zinc quantification. M.S. and A.G.F. assembled and mapped whole genome sequencing reads of suppressor strains. D.D. and S.A.-S. performed zebrafish zinc measurements. J.K., R.G., Y.L., and I.A. A. performed and analyzed RNA sequencing experiments in mouse and patient samples. C.G. and A.-C.G. performed proteomics and structure threading experiments. W.B.D. conceived the study, performed apoptosis experiments, and helped write the manuscript. All authors reviewed the manuscript before publication.

## Additional information

**Competing interests:** The authors declare no competing interests.

