## [Peer Review File · Nature Communications]

Reviewers' comments:

Reviewer #1 (Remarks to the Author):

Chapman et al. employ a combination of forward genetic and proteomic approaches in *C. elegans* to dissect cell non-autonomous mechanisms of CCM1/KRIT1 action in the apoptotic cell death in the worm intestine. The Derry laboratory has been using worm molecular genetics to investigate the biology of the KRI-1, the mammalian ortholog of which, CCM1/KRIT1, is one of three proteins mutated in familial (but also at least some sporadic) forms of Cerebral Cavernous Malformations, vascular lesions that develop in the central nervous system. The authors identify a zinc transporter as the mediator of pro-apoptotic action of the CCM complex and extend these observations to zebrafish and mouse models and human CCM lesions.

The work is in general very interesting, well executed, suggesting an intriguing and novel aspect of CCM1/KRIT1 function, and takes advantage of the powerful worm genetics. The *C. elegans* experiments conclusively demonstrate conservation of the CCM complex (via the identification of the up to now elusive worm CCM2 ortholog) and propose a mechanism by which it influences intestinal apoptosis involving zinc regulation. In contrast, I find the vertebrate data to be too preliminary, and not adding significantly to the work, as they invite more questions than they answer (in addition to being quite simplistic when contrasted with the data obtained in the worm). I appreciate that these, rather superficial, experiments were included in the study in an attempt to enhance the importance of the authors' main findings by suggesting they can be applicable in vertebrate organisms as well, but I would like to argue that the basic findings in the worm constitute themselves a well-rounded and self-contained story.

Specific comments are below:

Line 84/85: please explain "in a manner similar to KRI-1"; also, why would it necessarily act downstream?

Lines 107-112: there is a logical gap in this section – it might be better to state that an alternative to scoring apoptosis is needed, hence *kri-1* larvae were tested for response to starvation, which proved to be an effective screening method.

Line 116-118: Could you provide details about the remaining 7 suppressor strains?

Line 142/143: "strongly suggest..." is almost verbatim repeated in the title of the next sub-section; consider rephrasing. Also, this is a very emphatic statement – please moderate, because this conclusion is only supported by the experiments described in the following section (as is started in line 154).

Line 216/217 and 228/229: please clearly state that you propose K07A9.3 and F37C4.5 as the worm orthologs of CCM2 and ICAP1, respectively; the former is a significant finding and should be highlighted appropriately. However, as this is clarified at the end of the paragraph, please consider referring to the two genes simply as K07A9.3 and F37C4.5 in lines 216/217. The supporting data would even be presented as part of the main figures.

Line 238: please qualify "this"

Line 283: please specify "KRI-1-MPK-2-KLF-3 signalling cascade" in the soma

Line 339-341: please elaborate on the physiological role of zinc in zebrafish and discuss why it may be localized only to some parts of the vasculature.

Line 350: please clarify here that these are KRT1/CCM1 mutated lesions?

Minor comments:

Line 53: please add "ionizing" radiation (IR)-induced (or, does it stand for "irradiation?")

Lines 84/85: ref 10 should be grouped with ref 9 (not ref 12)

MPK1 is sometimes referred to as MPK1/ERK1; please adopt consistent terminology throughout the manuscript

Line 113: "restore" viability is awkward – extend survival?

Line 129: "shared" instead of "contained"?

Reviewer #2 (Remarks to the Author):

Derry and co-workers have previously shown that the *C. elegans* homolog of the mammalian scaffold protein KRIT1/CCM1, KRI-1, 'permits' in a cell non-autonomous manner, the apoptotic death of germ cells in response to DNA-damage. In this manuscript now they present evidence that KRI-1 CCM1 acts in a complex that also contains a *C. elegans* homolog of the MEKK3 scaffold protein CCM2, CCM-2 and that this complex controls the activity of the *C. elegans* ERK5 pathway, which in turn controls the expression of the transcription factor KLF-3. KLF-3 then controls the expression of the zinc transporter ZIPT-2.3, which is required for the sequestration of zinc in the intestine. In animals lacking *kri-1*, these interacting pathways and factors lead to the inability of animals to sequester zinc in the intestine and this results in increased zinc levels in the body cavity. Based on these findings the authors propose that this leads to increased levels of zinc in the germline and the inhibition of MPK-1, whose function is required for the apoptotic death of germ cells. Finally, the authors present some data that a disruption of zinc levels could also be involved in CCM pathology in zebrafish and mice.

This is an excellent follow-up to the authors' previous finding that the loss of CCM1 in *C. elegans* affects germ cell apoptosis non cell-autonomously and the authors present mostly convincing data that this is through changes in the level of zinc. In addition, the authors implicated altered levels of zinc in CCM in higher organisms as well. Therefore, this is a study that is important not only for our understanding of the control of germ cell apoptosis but of CCM. However, I have a few comments which should be addressed by the authors.

1. Using dyes that detect zinc, the authors present evidence that in *kri-1* mutants, the level of zinc is reduced in the intestine and increased in the body cavity. However, they do not demonstrate that this leads to increased levels of zinc in the germ line. If possible, this should be done. In addition, the germ line is surrounded by the somatic gonad. The level of zinc should therefore also be increased in this tissue. Have the authors looked at this? Finally, are there zinc transporters expressed on the PM of the somatic gonad or the germ line which might be necessary for the effect on germ cell apoptosis observed in *kri-1* animals?

2. Fluorescent images in Figures 1, 3, 5. Some of the images are not very convincing and should perhaps be quantified (especially Fig. 1a and Fig. 5c and d.)

3. Dot blots, instead of using different small black symbols for the different genotypes, maybe different colors should be used? It's sometimes not easy to see what dots belong to what genotype.

4. Gene names. The authors should give official gene names also to the components of the ERK5 pathway and use those names throughout the manuscript and all figures.

Reviewer #3 (Remarks to the Author):

This manuscript from Professor Derry and his team details the molecular mechanisms of IR-induced apoptosis in *C. elegans*, with a unique focus on the communication between somatic and germ cells, and reveals the importance of zinc within this response. The authors then broaden the findings to zebrafish and human cells from patients with cerebral cavernous malformations (CCMs). The

manuscript follows from Derry's findings reports in previous years on the role of CCMs in vesicle trafficking and kinase cascades, but the current manuscript is not redundant to that work. The manuscript has the added value in the identification of several worm homologs to mammalian genes and provides some insights into zinc homeostasis, at least in the worm. The manuscript also has relevance to the clinical understanding of CCM by proposing ERK5 signaling might contribute to the pathology of the disease. Overall, I feel that the quality of the research, breath of the topic, and connections to multiple sub-disciplines makes the manuscript worthy for publication in this journal. I would like to offer some major and minor suggestions for the authors to address.

Major:

- The zinc measurements of zinc within the worms depend fully on indirect fluorescent probe methods. While FluorZin and Zinpyrl are effective, they are not perfect and have some limits in their pharmacodynamics. It would be valuable to buttress the probe results with direct measures for zinc, like ICP-MS.
- Similarly, while helpful to know the concentration of exogenously added zinc, that doesn't provide information on bioavailability, so zinc content in the worms after treatment would be worth providing. Was toxicity and other phenotype assessed after exogenous zinc?
- The reduction of apoptosis with exogenous zinc after IR in wildtype (Fig4G) is a key finding but the effect was rather modest – I'm a bit surprised the change reaches significance. I understand that more zinc precipitated, but seems worth trying other approach to generate a dose-response for zinc and apoptosis. Also, how about using TPEN to reduce zinc levels? TPEN has been used effectively in worms.
- A schematic in the main figures, such as the one in supplemental figure 1, would be helpful to the reader to keep the complex pathway straight.

Minor:

- The Discussion could benefit from more detail on how zinc homeostasis in worms effects on lifespan and toxicity, since the effect of zinc on sensitivity for apoptosis triggers is in a broader context of zinc effects on worm physiology. Authors are referred to DOIs 10.1016/j.etap.2007.05.009, 10.1371/journal.pgen.1002013, & 10.1371/journal.pone.0153513.
- CCM should be defined in the abstract, since it may not be known by a broad reading audience.
- Cell-non/autonomous action should be briefly defined in the abstract or introduction, since it may not be known by a broad reading audience.
- It's not clear what the authors mean by "It is thus possible that high levels of zinc in the body cavity of kri-1(ok1251) mutants results in the permeabilization."
- In the section heading of results, the text sometimes says just promotion of apoptosis. It should always stay "radiation-induced apoptosis" since different apoptosis stimuli have different mechanisms of activation.
- Small inconsistencies in syntax, e.g. dash inclusion in CCMx notations, space/no space between units and numerals in concentrations, space/no space before reference notations, etc.

Below are the comments from each reviewer and our responses in blue.

Reviewer #1 (Remarks to the Author):

Chapman et al. employ a combination of forward genetic and proteomic approaches in *C. elegans* to dissect cell non-autonomous mechanisms of CCM1/KRIT1 action in the apoptotic cell death in the worm intestine. The Derry laboratory has been using worm molecular genetics to investigate the biology of the KRI-1, the mammalian ortholog of which, CCM1/KRIT1, is one of three proteins mutated in familial (but also at least some sporadic) forms of Cerebral Cavernous Malformations, vascular lesions that develop in the central nervous system. The authors identify a zinc transporter as the mediator of pro-apoptotic action of the CCM complex and extend these observations to zebrafish and mouse models and human CCM lesions.

The work is in general very interesting, well executed, suggesting an intriguing and novel aspect of CCM1/KRIT1 function, and takes advantage of the powerful worm genetics. The *C. elegans* experiments conclusively demonstrate conservation of the CCM complex (via the identification of the up to now elusive worm CCM2 ortholog) and propose a mechanism by which it influences intestinal apoptosis involving zinc regulation. In contrast, I find the vertebrate data to be too preliminary, and not adding significantly to the work, as they invite more questions than they answer (in addition to being quite simplistic when contrasted with the data obtained in the worm). I appreciate that these, rather superficial, experiments were included in the study in an attempt to enhance the importance of the authors' main findings by suggesting they can be applicable in vertebrate organisms as well, but I would like to argue that the basic findings in the worm constitute themselves a well-rounded and self-contained story.

We appreciate the suggestion that the worm data constitute a self-contained story. However, while preliminary, we feel that the vertebrate and human data demonstrate the conservation of zinc localization by CCM proteins throughout evolution. We indicate in the main body of the text that this data is indeed preliminary and hope you and the reviewers agree that it is appropriate to leave this data in the manuscript.

Specific comments are below:

Line 84/85 (previous version): please explain “in a manner similar to KRI-1”; also, why would it necessarily act downstream?

Sentence modified to say the following: “Given that the ERK1 homologue, MPK-1, in the pachytene region of the germline is necessary for IR-induced apoptosis^{9,10} we first wondered if germline MPK-1 might be regulated by KRI-1.” (Line 86/87)

Lines 107-112 (previous version): there is a logical gap in this section – it might be better to state that an alternative to scoring apoptosis is needed, hence *kri-1* larvae were tested for response to starvation, which proved to be an effective screening method.

Sentence modified to say the following: “To identify genes that function downstream of kri-1, we performed a forward genetic suppressor screen using the mutagen ethyl methanesulfonate (EMS). Since scoring apoptosis in single worms under a compound light microscope is rate-limiting, a rapid screening method was required. We therefore exploited the hypersensitivity of kri-1(ok1251) mutants to starvation stress to select for EMS-induced mutations that suppress this kri-1 phenotype (Figure 2A), reasoning that selecting for such mutations might also restore IR-induced germline apoptosis.” (Line 110-115)

Line 116-118 (previous version): Could you provide details about the remaining 7 suppressor strains?

Sentence changed to clarify that only 13 of the kir-1(ok1251) suppressor strains had a full restoration of IR-induced apoptosis: “Clonal populations from these survivors were established, and 13 of these suppressor strains isolated from unique populations had a complete restoration of IR-induced apoptosis (Figure 2B).” (Line 118-120)

Line 142/143: “strongly suggest...” is almost verbatim repeated in the title of the next subsection; consider rephrasing. Also, this is a very emphatic statement – please moderate, because this conclusion is only supported by the experiments described in the following section (as is started in line 154).

Sentence modified to say the following: “Since the vertebrate ERK5 pathway is known to regulate KLF transcription factors in the context of CCM disease, it is possible that klf-3 is part of the KRI-1/KRIT1 pathway in C. elegans.” (Line 140-142)

Line 216/217 and 228/229: please clearly state that you propose K07A9.3 and F37C4.5 as the worm orthologs of CCM2 and ICAP1, respectively; the former is a significant finding and should be highlighted appropriately. However, as this is clarified at the end of the paragraph, please consider referring to the two genes simply as K07A9.3 and F37C4.5 in lines 216/217. The supporting data would even be presented as part of the main figures.

Stated earlier in the section that we propose K07A9.3 and F37C4.5 to be the worm orthologs of CCM2 and ICAP1: “Due to their structural homology, we henceforth refer to K07A9.3 as ccm-2 and Y45F10D.10 as icap-1.” (Line 216/217)

Line 238: please qualify “this”

We removed the following sentence from the updated manuscript because it is superfluous: “This is reminiscent of changes observed in transcription factor mutants and indicates that transcription is regulated downstream of KRI-1.”

Line 283: please specify “KRI-1-MPK-2-KLF-3 signalling cascade” in the soma

Added “in the soma” to the sentence: “Collectively, these results indicate that the KRI-1-MPK-2-KLF-3 signalling cascade in the soma ensures proper storage of zinc in the intestine.” (Line 276/277)

Line 339-341: please elaborate on the physiological role of zinc in zebrafish and discuss why it may be localized only to some parts of the vasculature.

Added additional text to the discussion: “In zebrafish, zinc is known to regulate brain acetylcholine levels, memory and motor activities, while zinc chelation results in neuroprotective effects in a zebrafish model of hypoxic brain injury. These findings reveal that the regulation of zinc levels might be important for normal brain function, and possibly to prevent the development of neurological diseases such as CCM.” (Line 474-478)

Line 350: please clarify here that these are KRT1/CCM1 mutated lesions?

Added “four sporadic lesions and one familial CCM1 lesion” to the sentence: “To determine if the transcriptional regulation of SLC39 zinc transporters is relevant to the pathology of CCM disease, we analyzed endothelial cell (EC) RNA from five human CCM lesions (four sporadic and one familial CCM1) and compared these samples to corresponding brain tissue from three patients free of neurological disease.” (Line 385-388)

Minor comments:

Line 53: please add “ionizing” radiation (IR)-induced (or, does it stand for “irradiation?)

“ionizing radiation” added to the sentence: “In addition, the ERK1/2 homologue MPK-1 cooperates with the CEP-1 pathway and can also function in parallel to promote ionizing radiation (IR)-induced apoptosis.” (Line 51-53)

Lines 84/85: ref 10 should be grouped with ref 9 (not ref 12)

Complete: “Given that the ERK1 homologue, MPK-1, in the pachytene region of the germline is necessary for IR-induced apoptosis^{9,10} we first wondered if germline MPK-1 might be regulated by KRI-1.” (Line 86/87)

MPK1 is sometimes referred to as MPK1/ERK1; please adopt consistent terminology throughout the manuscript

“MPK-1” is defined as the ERK1 homologue (Line 86) and MPK-1 is now used consistently throughout the text.

Line 113: “restore” viability is awkward – extend survival?

Reworded the sentence: “We therefore exploited the hypersensitivity of kri-1(ok1251) mutants to starvation stress to select for EMS-induced mutations that suppress this kri-1 phenotype (Figure 2A), reasoning that selecting for such mutations might also restore IR-induced germline apoptosis. (Line 113-115)

Line 129: “shared” instead of “contained”?

Changed: “Of these six strains, two had unique point mutations, while four strains shared a third point mutation (Figure 2C).” (Line 128/129)

Reviewer #2 (Remarks to the Author):

Derry and co-workers have previously shown that the *C. elegans* homolog of the mammalian scaffold protein KRIT1/CCM1, KRI-1, 'permits' in a cell non-autonomous manner, the apoptotic death of germ cells in response to DNA-damage. In this manuscript now they present evidence that KRI-1 CCM1 acts in a complex that also contains a *C. elegans* homolog of the MEKK3 scaffold protein CCM2, CCM-2 and that this complex controls the activity of the *C. elegans* ERK5 pathway, which in turn controls the expression of the transcription factor KLF-3. KLF-3 then controls the expression of the zinc transporter ZIPT-2.3, which is required for the sequestration of zinc in the intestine. In animals lacking *kri-1*, these interacting pathways and factors lead to the inability of animals to sequester zinc in the intestine and this results in increased zinc levels in the body cavity. Based on these findings the authors propose that this leads to increased levels of zinc in the germline and the inhibition of MPK-1, whose function is required for the apoptotic death of germ cells. Finally, the authors present some data that a disruption of zinc levels could also be involved in CCM pathology in zebrafish and mice.

This is an excellent follow-up to the authors' previous finding that the loss of CCM1 in *C. elegans* affects germ cell apoptosis non cell-autonomously and the authors present mostly convincing data that this is through changes in the level of zinc. In addition, the authors implicated altered levels of zinc in CCM in higher organisms as well. Therefore, this is a study that is important not only for our understanding of the control of germ cell apoptosis but of CCM. However, I have a few comments which should be addressed by the authors.

1. Using dyes that detect zinc, the authors present evidence that in *kri-1* mutants, the level of zinc is reduced in the intestine and increased in the body cavity. However, they do not demonstrate that this leads to increased levels of zinc in the germ line. If possible, this should be done. In addition, the germ line is surrounded by the somatic gonad. The level of zinc should therefore also be increased in this tissue. Have the authors looked at this?

As an attempt determine if levels of zinc increase in the germline or somatic gonad, we dissected gonads from worms grown on various zinc concentrations and tried to detect zinc in these isolated gonads using Zinpyr-1 and FluoZin-3. Unfortunately, we were unable to detect any signal in the germline or somatic gonad using this method. In addition, we tried to inject these two dyes in germlines of intact animals but were unable to detect any signal. Finally, we tried to send over 100 isolated gonads for zinc quantification by Inductively Coupled Plasma-Mass Spectrometry (ICP-MS), however, these were not enough to achieve a detectable zinc reading (to detect zinc in whole worms, over 5,000 animals were sent for ICP-MS). This new data is presented in Figure S6E of the revised manuscript.

Finally, are there zinc transporters expressed on the PM of the somatic gonad or the germ line which might be necessary for the effect on germ cell apoptosis observed in *kri-1* animals?

Currently, only zipt-7.1 is known to be expressed in the germline (Zhao et al. 2018. DOI: 10.1371/journal.pbio.2005069). To determine if zipt-7.1 has a role in IR-induced apoptosis, we knocked it down in wild type and kri-1(ok1251) animals but observed no effect in either strain (Figure S5D in the revised manuscript). We therefore performed an RNAi screen on 19/26 of the remaining zinc transporters available in our RNAi library to determine if knockdown of any gene results in apoptotic defects. If so, we could make reporters of such candidates to determine potential germline or somatic gonad expression. Unfortunately, knockdown of these genes in kri-1(ok1251) mutants failed to restore IR-induced apoptosis, suggesting that none are responsible for the suppression of IR-induced apoptosis in kri-1(ok1251) mutants (Figure S5E in the revised manuscript).

2. Fluorescent images in Figures 1, 3, 5. Some of the images are not very convincing and should perhaps be quantified (especially Fig. 1a and Fig. 5c and d.)

ImageJ quantifications added to Figure 1A, 3, 5 and S5A.

3. Dot blots, instead of using different small black symbols for the different genotypes, maybe different colors should be used? It's sometimes not easy to see what dots belong to what genotype.

All figures were changed to include enlarged symbols and added colours to every dot blot graph throughout the manuscript.

4. Gene names. The authors should give official gene names also to the components of the ERK5 pathway and use those names throughout the manuscript and all figures.

Y016G6A.1 is referred to as mekk-3 and E02D9.1 is referred to as mek-5 throughout the text. Also, K07A9.3 and Y45F10D.10 are now referred to ccm-2 and icap-1, respectively.

Reviewer #3 (Remarks to the Author):

This manuscript from Professor Derry and his team details the molecular mechanisms of IR-induced apoptosis in *C. elegans*, with a unique focus on the communication between somatic and germ cells, and reveals the importance of zinc within this response. The authors then broaden the findings to zebrafish and human cells from patients with cerebral cavernous malformations (CCMs). The manuscript follows from Derry's findings reports in previous years on the role of CCMs in vesicle trafficking and kinase cascades, but the current manuscript is not redundant to that work. The manuscript has the added value in the identification of several worm homologs to mammalian genes and provides some insights into zinc homeostasis, at least in the worm. The manuscript also has relevance to the clinical understanding of CCM by proposing ERK5 signaling might contribute to the pathology of the disease. Overall, I feel that the quality of the

research, breath of the topic, and connections to multiple sub-disciplines makes the manuscript worthy for publication in this journal. I would like to offer some major and minor suggestions for the authors to address.

Major:

- The zinc measurements of zinc within the worms depend fully on indirect fluorescent probe methods. While FluorZin and Zinpyrl are effective, they are not perfect and have some limits in their pharmacodynamics. It would be valuable to buttress the probe results with direct measures for zinc, like ICP-MS.

ICP-MS was optimized to quantify zinc in 5,000 whole worms, using the protocol described in Davis et al. 2009 (DOI: 10.1534/genetics.109.103614). This method revealed that in the absence of exogenous zinc, there is no difference in total zinc across the strains tested (Figure S6E in the revised manuscript).

- Similarly, while helpful to know the concentration of exogenously added zinc, that doesn't provide information on bioavailability, so zinc content in the worms after treatment would be worth providing.

*Similar to above, ICP-MS was optimized for 5,000 whole worms, following the protocol of Davis et al. 2009 (DOI: 10.1534/genetics.109.103614). Total zinc levels were higher in all strains after incubation with zinc, confirming the bioavailability of exogenous zinc (Figure S6E). Total zinc (after exogenous treatment) was lower in *kri-1(ok1251)* mutants, compared to wild type and *kri-1(ok1251); mpk-2(219)* animals, revealing the inability of *kri-1(ok1251)* animals to store zinc (Figure S6E in the revised manuscript).*

- Was toxicity and other phenotypes assessed after exogenous zinc?

We do not observe growth defects, egg laying defects, or fertility defects at the concentrations of zinc tested in this manuscript.

- The reduction of apoptosis with exogenous zinc after IR in wildtype (Fig4G) is a key finding but the effect was rather modest – I'm a bit surprised the change reaches significance. I understand that more zinc precipitated, but seems worth trying other approach to generate a dose-response for zinc and apoptosis.

As an alternative approach, we incubated worms on media with exogenous zinc for a short period of time (from L4 stage), as opposed to growing worms on zinc from the L1 stage in the previous version of the manuscript. We found that this method resulted in the suppression of IR-induced apoptosis at 0.1mM, 0.5mM, and 1mM of exogenous zinc (Figure S6A), as opposed to 1mM of zinc required to reach significance (and 0.5mM barely reaching significance) in the previous version of the manuscript.

We also incubated worms in tubes with M9 buffer supplemented with 0.1mM zinc for 4 hours and found that this method also resulted in a suppression of IR-induced apoptosis (Figure S6B).

- Also, how about using TPEN to reduce zinc levels? TPEN has been used effectively in worms.

We found that incubation of worms in tubes with liquid M9 buffer and 25 μ M TPEN for 4hrs increased IR-induced apoptosis in wild type animals, and restored IR-induced apoptosis in kri-1(ok1251) mutants (Figure S6C). We also confirmed using the zinc dye Zinpyr-1 that this TPEN treatment reduces zinc in the body cavity (Figure S6D).

- A schematic in the main figures, such as the one in supplemental figure 1, would be helpful to the reader to keep the complex pathway straight.

Switched figure S1B with figure 1C from the original manuscript and updated figure numbers throughout the updated manuscript to reflect this change.

Minor:

- The Discussion could benefit from more detail on how zinc homeostasis in worms effects on lifespan and toxicity, since the effect of zinc on sensitivity for apoptosis triggers is in a broader context of zinc effects on worm physiology. Authors are referred to DOIs 10.1016/j.etap.2007.05.009, 10.1371/journal.pgen.1002013, & 10.1371/journal.pone.0153513.

*Section added to the discussion of the updated manuscript on how zinc homeostasis in worms affects lifespan and toxicity and referenced the three suggested papers: “Zinc is required for the normal functions of about 3000 proteins, indicating that this cation is very important in cellular and developmental biology⁵⁶. This is apparent from increased levels of zinc resulting in lifespan, developmental, reproductive, locomotive, and chemotaxis defects⁵⁷ in *C. elegans*. Furthermore, zinc exposure was shown to activate stress responses in most tissues, and these behavioral toxicities were found to be transferred to progeny⁵⁷. Thus, *C. elegans* employs strict control to maintain zinc homeostasis, in order to prevent excess zinc from inducing adverse effects such as preventing stress-induced apoptosis. For instance, the histidine ammonia lyase, HALY-1, regulates zinc tolerance by modulating levels of histidine⁵⁸. Since histidine functions as a zinc chelator, higher levels of this amino acid can reduce the toxic effects of excess zinc⁵⁸. Inversely, reducing zinc levels in *C. elegans* with zinc chelators results in increased lifespan⁵⁹, and lower zinc levels are correlated with improved proteostasis⁵⁹. Collectively, this indicates that zinc levels are critical for maintaining cellular health, which we also found to be true with regards to apoptosis regulation.” (Line 438-451)*

- CCM should be defined in the abstract, since it may not be known by a broad reading audience.

Done: “...Cerebral Cavernous Malformations (CCM) patient tissues containing endothelial cells.” (Line 37)

- Cell-non/autonomous action should be briefly defined in the abstract or introduction, since it may not be known by a broad reading audience.

“Cell autonomous” and “non-cell autonomous” are now defined in the introduction of the updated manuscript: “While stress-induced apoptosis was originally thought to be activated from within the germline as a cell-autonomous process, we previously found that KRI-1 is required in the soma to promote IR-induced germline apoptosis in a cell non-autonomous manner, independent of CEP-1/p53¹².” (Lines 55-57)

- It's not clear what the authors mean by “It is thus possible that high levels of zinc in the body cavity of kri-1(ok1251) mutants results in the permeabilization.”

Modified this sentence for clarity in the updated manuscript: “It is possible that a reduction of zinc storage in the intestine of kri-1(ok1251) mutants results in higher levels of zinc in the body cavity of these animals.” (Lines 287/288)

- In the section heading of results, the text sometimes says just promotion of apoptosis. It should always stay “radiation-induced apoptosis” since different apoptosis stimuli have different mechanisms of activation.

In the updated version of the manuscript we now consistently say “IR-induced apoptosis” throughout the text.

- Small inconsistencies in syntax, e.g. dash inclusion in CCMx notations, space/no space between units and numerals in concentrations, space/no space before reference notations, etc.

The updated manuscript is now consistent in the following:

- 1) Dash inclusion in gene and protein names for C. elegans, no dash for vertebrates.*
- 2) Removed spaces between numerals and units.*
- 3) Removed spaces before reference notations.*

REVIEWERS' COMMENTS:

Reviewer #1 (Remarks to the Author):

The authors have addressed the points raised in my review in a comprehensive and satisfactory manner.

Reviewer #2 (Remarks to the Author):

Derry and co-workers have addressed some of the questions and concerns I had about the previous version of the manuscript and I acknowledge that some of them cannot be addressed at the moment for technical reasons (determination of zinc level in the germline for example). The revised manuscript is overall improved.

Reviewer #3 (Remarks to the Author):

This manuscript from Professor Derry and his team has responded successfully to all points raised by this Reviewer. The revised manuscript reads well and is ready for publication.